# Break It Down: Evidence for Structural Compositionality in Neural Networks

**Michael A. Lepori**[1][*]    **Thomas Serre**[2]    **Ellie Pavlick**[1]
[1]Department of Computer Science    [2]Carney Institute for Brain Science
Brown University

## Abstract

Though modern neural networks have achieved impressive performance in both vision and language tasks, we know little about the functions that they implement. One possibility is that neural networks implicitly break down complex tasks into subroutines, implement modular solutions to these subroutines, and compose them into an overall solution to a task — a property we term *structural compositionality*. Another possibility is that they may simply learn to match new inputs to learned templates, eliding task decomposition entirely. Here, we leverage model pruning techniques to investigate this question in both vision and language across a variety of architectures, tasks, and pretraining regimens. Our results demonstrate that models often implement solutions to subroutines via modular subnetworks, which can be ablated while maintaining the functionality of other subnetworks. This suggests that neural networks may be able to learn compositionality, obviating the need for specialized symbolic mechanisms.

## 1   Introduction

Though neural networks have come to dominate most subfields of AI, much remains unknown about the functions that they learn to implement. In particular, there is debate over the role of *compositionality*. Compositionality has long been touted as a key property of human cognition, enabling humans to exhibit flexible and abstract language processing and visual processing, among other cognitive processes (Marcus, 2003; Piantadosi et al., 2016; Lake et al., 2017; Smolensky et al., 2022). According to common definitions (Quilty-Dunn et al., 2022; Fodor & Lepore, 2002), a representational system is compositional if it implements a set of discrete constituent functions that exhibit some degree of modularity. That is, *blue circle* is represented compositionally if a system is able to entertain the concept *blue* independently of *circle*, and vice-versa.

It is an open question whether neural networks require explicit symbolic mechanisms to implement compositional solutions, or whether they implicitly learn to implement compositional solutions during training. Historically, artificial neural networks have been considered non-compositional systems, instead solving tasks by matching new inputs to learned templates (Marcus, 2003; Quilty-Dunn et al., 2022). Neural networks' apparent lack of compositionality has served as a key point in favor of integrating explicit symbolic mechanisms into contemporary artificial intelligence systems (Andreas et al., 2016; Koh et al., 2020; Ellis et al., 2023; Lake et al., 2017). However, modern neural networks, with no explicit inductive bias towards compositionality, have demonstrated successes on increasingly complex tasks. This raises the question: are these models succeeding by implementing compositional solutions under the hood (Mandelbaum et al., 2022)?

---

[*]Correspondence to: `michael_lepori@brown.edu`

37th Conference on Neural Information Processing Systems (NeurIPS 2023).

**Contributions and Novelty:**

1. We introduce the concept of *structural compositionality*, which characterizes the extent to which neural networks decompose compositional tasks into subroutines and implement them modularly. We test for structural compositionality in several different models across both language and vision[2].

2. We discover that, surprisingly, there is substantial evidence that many models implement subroutines in modular subnetworks, though most do not exhibit perfect task decomposition.

3. We characterize the effect of unsupervised pretraining on structural compositionality in fine-tuned networks and find that pretraining leads to a more consistently compositional structure in language models.

This study contributes to the emerging body of work on "mechanistic interpretability" (Olah, 2022; Cammarata et al., 2020; Ganguli et al., 2021; Henighan et al., 2023) which seeks to explain the algorithms that neural networks implicitly implement within their weights. We make use of techniques from model pruning in order to gain insight into these algorithms. While earlier versions of these techniques have been applied to study modularity in a multitask setting (Csordás et al., 2021), our work is novel in that it applies the method to more complex language and vision models, studies more complex compositional tasks, and connects the results to a broader discussion about defining and measuring compositionality within neural networks.

## 2    Structural Compositionality

Most prior work on compositionality in neural networks has focused on whether they *generalize* in accordance with the compositional properties of data (Ettinger et al., 2018; Kim & Linzen, 2020; Hupkes et al., 2020). Such work has mostly yielded negative results – i.e., evidence that neural networks fail to generalize compositionally. This work is important for understanding how current models will behave in practice. However, generalization studies alone permit only limited conclusions about how models work.

As discussed above, leading definitions of compositionality are defined in terms of a system's representations, not its behavior. That is, definitions contrast compositional systems (which implement modular constituents) with noncompositional systems (which might, e.g., rely on learned templates). Poor performance on generalization studies does not differentiate these two types of systems, since even a definitionally compositional system might fail at these generalization tasks. For example, a Bayesian network that explicitly represents and composes distinct shape and color properties might nonetheless classify a *blue circle* as a *red circle* if it has a low prior for predicting the color blue and a high prior for predicting the color red.

Thus, in this work, we focus on evaluating the extent to which a model's representations are *structured* compositionally. Consider the task described in Figure 1. In this task, a network learns to select the "odd-one-out" among four images. Three of them follow a compositional rule (they all contain two shapes, one of which is **inside** and **in contact** with the other). One of them breaks this rule. There are at least two ways that a network might learn to solve this type of compositional task. (1) A network might compare new inputs to prototypes or iconic representations of previously-seen inputs, avoiding any decomposition of these prototypes into constituent parts (i.e., it might implement a *non-compositional solution*). (2) A network might implicitly break the task down into subroutines, implement solutions to each, and compose these results into a solution (i.e., it might implement a *compositional solution*). In this case, the subroutines consist of a **(+/- Inside)** detector and a **(+/- Contact)** detector.

If a model trained on this task exhibits *structural compositionality*, then we would expect to find a subnetwork that implements each subroutine within the parameters of that model. This subnetwork should compute one subroutine, and not the other (Figure 1, Bottom Right; "Subnetwork"), and it should be *modular* with respect to the rest of the network — it should be possible to ablate this subnetwork, harming the model's ability to compute one subroutine while leaving the other subroutine largely intact (Figure 1, Bottom Right; "Ablation"). However, if a model does not exhibit structural compositionality, then it has only learned the *conjunction* of the subroutines rather than

---

[2]Our code is publicly available at `https://github.com/mlepori1/Compositional_Subnetworks`.

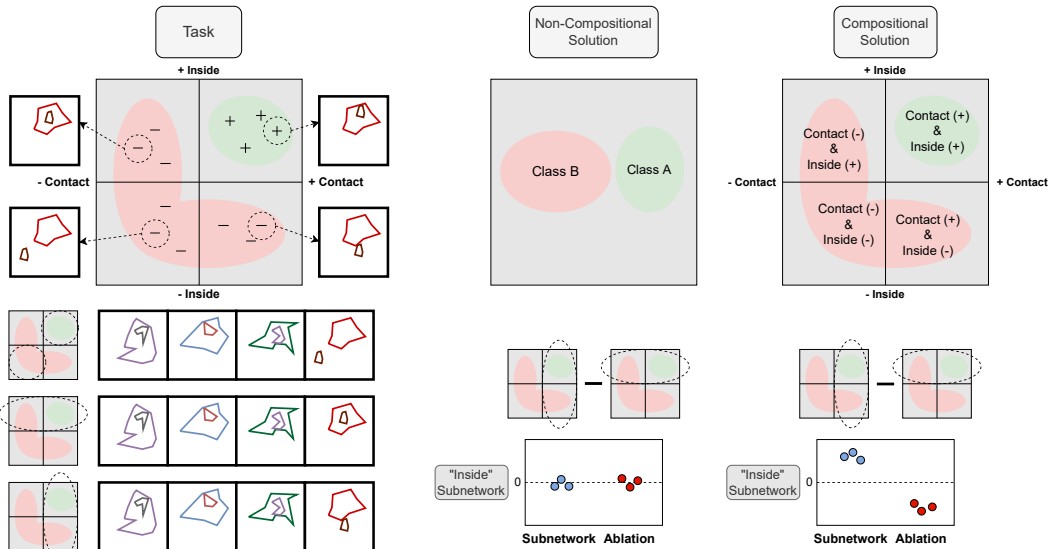

Figure 1: **(Left)** An illustration of the tasks used to study structural compositionality. Stimuli are generated via the composition of two subroutines: **(+/- Inside)** and **(+/- Contact)**. These stimuli are used to construct **odd-one-out** tasks, where the model is tasked with identifying the image that does not follow a rule from a set of four samples. Here, two objects must be in contact, and one must be inside the other. Rule following images correspond to the upper right quadrant. A model may solve this task in two ways. **(Middle)** It may implement a non-compositional solution, e.g., storing learned template that encodes only the conjunction of the two subroutines. In this case, one should not be able to find a subnetwork that implements one subroutine and does not implement the other. Concretely, there should be no difference in the subnetwork's performance on examples that depend on computing one subroutine vs. another. Ablating this subnetwork should harm the computation of both subroutines equally. In other words, there should be no difference in accuracy between examples that depend on different subroutines. **(Right)** A model may implement a compositional solution, which computes each subroutine in modular subnetworks and combines them. In this case, one should find a subnetwork that implements, say, **(+/- Inside)**, and this subnetwork should achieve high accuracy on examples that require computing **(+/- Inside)** and low performance on examples that require computing **(+/- Contact)**. In other words, the difference in accuracies between the **(+/- Inside)** and **(+/- Contact)** examples should be positive. Likewise, one should be able to ablate this subnetwork and maintain performance on **(+/- Contact)** while compromising performance on **(+/- Inside)**, and so the difference in performance should be negative. Hypothetical results are represented as differences in performance between both types of examples.

their *composition*. It should not be possible to find a subnetwork that implements one subroutine and not the other, and ablating one subnetwork should hurt accuracy on both subroutines equally (Figure 1, Bottom Center). This definition is related to prior work on modularity in neural networks (Csordás et al., 2021; Hod et al., 2022), but here we specifically focus on modular representations of compositional tasks.

## 3 Experimental Design

### 3.1 Preliminaries

Here we define terms used in the rest of the paper. **Subroutine:** A binary rule. The $i^{th}$ subroutine is denoted $SR_i$. **Compositional Rule:** A binary rule that maps input to output according to $C = SR_1 \& SR_2$, where $SR_i$ is a subroutine. Compositional rules are denoted $C$. **Base Model:** A model that is trained to solve a task defined by a compositional rule. Denoted $M_C$. **Subnetwork:** A subset of the parameters of a base model, which implements one subroutine. The subnetwork that implements $SR_i$ is denoted $Sub_i$. This is implemented as a binary mask, $m_i$, over the parameters of the base model, $\theta$, such that $Sub_i = M_{C;\theta \odot m_i}$, where $\odot$ refers to elementwise multiplication.

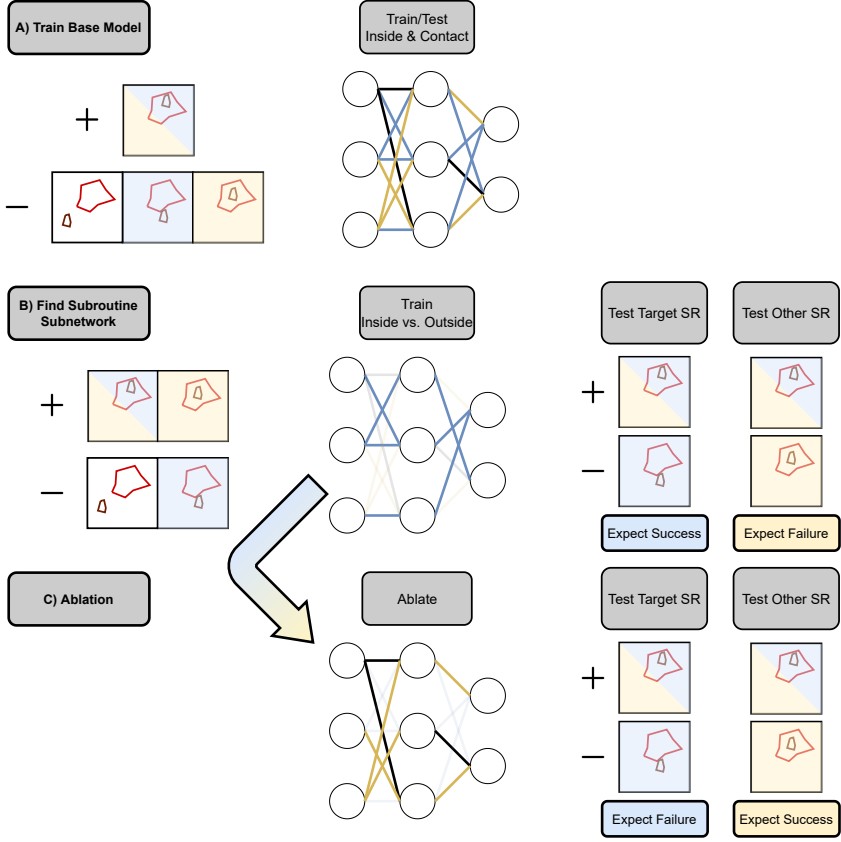

Figure 2: Illustration of the experimental design. For brevity, we denote "subroutine" as **SR** in the diagram. **(A)** First, we train a neural network on a compositional task **(Inside-Contact)**, ensuring that it can achieve high accuracy on the task. **(B)** We then optimize a binary mask over weights, such that the resulting subnetwork can compute one subroutine **(+/- Inside)** while ignoring the other **(+/- Contact)**. We evaluate this subnetwork on datasets that require computing the target subroutine **(+/- Inside)**. We also evaluate this subnetwork on datasets that require computing the other subroutine **(+/- Contact)**. We expect success on the first evaluation and failure on the second if the model exhibits structural compositionality. **(C)** We invert the binary mask learned in **(B)**, ablating the subnetwork. We evaluate this on the same two datasets, expecting performance to be harmed on the target subroutine and performance to be high for the other subroutine.

**Ablated Model:** The complement set of parameters of a particular subnetwork. After ablating $Sub_i$, we denote the ablated model $M_{ablate_i}$.

## 3.2 Experimental Logic

Consider a compositional rule, $C$, such as the "Inside-Contact" rule described in Figures 1 and 2. The rule is composed of two subroutines, $SR_1$ **(+/- Inside)** and $SR_2$ **(+/- Contact)**. We define an odd-one-out task on $C$, as described in Section 2. See Figure 1 for three demonstrative examples using the "Inside-Contact" compositional rule. For a given architecture and compositional rule, $C$, we train a base model, $M_C$, such that $M_C$ solves the odd-one-out task to greater than 90% accuracy[3] (Figure 2, Panel A). We wish to characterize the extent to which $M_C$ exhibits structural compositionality. Does $M_C$ learn only the conjunction (effectively entangling the two subroutines), or does $M_C$ implement $SR_1$ and $SR_2$ in modular subnetworks?

---

[3]This threshold was selected arbitrarily, but our results do not depend on it. All models end up achieving $>$ 99% accuracy (See Appendix A).

To investigate this question, we will learn a binary mask $m_i$ over the weights $\theta$ of $M_C$ for each $SR_i$, resulting in a subnetwork $Sub_i$. Without loss of generality, assume $Sub_1$ computes (**+/- Inside**) and $Sub_2$ computes (**+/- Contact**), and consider investigating $Sub_1$. We can evaluate this subnetwork on two partitions of the training set: (1) **Test Target Subroutine** – Cases where a model must compute the target subroutine to determine the odd-one-out (e.g., cases where an image exhibits (**- Inside, + Contact**)) and (2) **Test Other Subroutine** – Cases where a model must compute the other subroutine to determine the odd-one-out (e.g., cases where the odd-one-out exhibits (**+ Inside, - Contact**)).

Following prior work (Csordás et al., 2021), we assess structural compositionality based on the sub-network's performance on these datasets, as well as the base model's performance after *ablating* the subnetwork[4]. If $M_C$ exhibits structural compositionality, then $Sub_1$ should only be able to compute the target subroutine (**+/- Inside**), and thus it should perform better on **Test Target Subroutine** than on **Test Other Subroutine**. If $M_C$ entangles the subroutines, then $Sub_1$ will implement both subroutines and will perform equally on both partitions. See Figure 2, Panel B.

To determine modularity, we ablate the $Sub_1$ from the base model and observe the behavior of the resulting model, $M_{ablate_1}$. If $M_C$ exhibits structural compositionality, we expect the two subroutines to be modular, such that ablating $Sub_1$ has more impact on $M_{ablate_1}$'s ability to compute (**+/- Inside**) than (**+/- Contact**). Thus, we would expect $M_{ablate_1}$ to perform better on **Test Other Subroutine** than **Test Target Subroutine**. See Figure 2, Panel C. However, if $M_C$ implemented a non-compositional solution, then ablating $Sub_1$ should hurt performance on both partitions equally, as the two subroutines are entangled. Thus, performance on both partitions would be approximately equal.

**Expected Results:** For each model and task, our main results are the differences in performance between **Test Target Subroutine** and **Test Other Subroutine** for each subnetwork and ablated model. If a model exhibits structural compositionality, we expect the subnetwork to produce a positive difference in performance (**Test Target Subroutine** > **Test Other Subroutine**), and the corresponding ablated model to produce a negative difference in performance. Otherwise, we expect no differences in performance. See Figure 1 for hypothetical results.

## 4   Discovering Subnetworks

Consider a frozen model $M_C(\cdot; w)$ trained on an odd-one-out task defined using the compositional rule $C$. Within the weights of this model, we wish to discover a subnetwork that implements $SR_i$[5]. We further require that the discovered subnetwork should be as small as possible, such that if the model exhibits structural compositionality, it can be ablated with little damage to the remainder of the network. Thus, we wish to learn a binary mask over the weights of a trained neural network while employing $L_0$ regularization

Most prior work that relies on learning binary masks over network parameters (Cao et al., 2021; Csordás et al., 2021; Zhang et al., 2021; Guo et al., 2021; De Cao et al., 2020, 2022) relies on stochastic approaches, introduced in Louizos et al. (2018). Savarese et al. (2020) introduced *continuous sparsification* as a deterministic alternative to these stochastic approaches and demonstrated that it achieves superior pruning performance, both in terms of sparsity and subnetwork performance. Thus, we use continuous sparsification to discover subnetworks within our models. See Appendix B for details.

## 5   Vision Experiments

**Tasks:** We extend the collection of datasets introduced in Zerroug et al. (2022), generating several tightly controlled datasets that implement compositions of the following subroutines: **contact**, **inside**, and **number**. From these three basic subroutines, we define three compositional rules: **Inside-Contact**, **Number-Contact**, and **Inside-Number**. We will describe the **Inside-Contact** tasks in detail, as the same principles apply to the other two compositional rules (See Appendix E). This

---

[4]This is similar to Csordás et al. (2021)'s $P_{\text{Specialize}}$ metric.

[5]Over all networks, we only mask weight parameters, leaving bias parameters untouched.

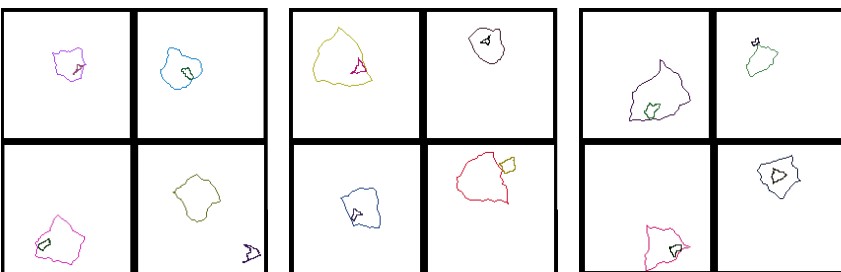

Figure 3: Three **Inside-Contact** stimuli. The odd-one-out is always the bottom-right image in these examples. **(Right)** An example from the task used to train the base model. **(Middle)** An example from the task used to discover the **+/- Inside** Subroutine. **(Left)** An example from the task used to discover the **+/- Contact** Subroutine.

task contains four types of images, each containing two shapes. In these images, one shape is either inside and in contact with the other **(+ Inside, + Contact)**, not inside of but in contact with the other **(- Inside, + Contact)**, inside of but not in contact with the other **(+ Inside, - Contact)**, or neither **(- Inside, - Contact)**. An example is defined as a collection of four images, three of which follow a rule and one of which does not. We train our base model to predict the odd-one-out on a task defined by a compositional rule over contact and inside: images of the type **(+ Inside, + Contact)** follow the rule, and any other image type is considered the odd-one-out. See Figure 3 (Left).

In order to discover a subnetwork that implements each subroutine, we define one odd-one-out task per subroutine. To discover the **+/- Inside** Subroutine, we define **(+ Inside)** to be rule-following (irrespective of contact) and **(- Inside)** to be the odd-one-out. Similarly for the **+/- Contact** Subroutine. See Figure 3 (Middle and Right, respectively). The base model has only seen data where **(+ Inside, + Contact)** images are rule-following. In order to align our evaluations with the base model's training data, we create two more datasets that probe each subroutine. For both, all rule-following images are **(+ Inside, + Contact)**. To probe for **(+/- Inside)**, the odd-one-out for one dataset is always a **(- Inside, + Contact)** image. This dataset is **Test Target Subroutine**, with respect to the subnetwork that implements **(+/- Inside)**. Similarly, to probe for **(+/- Contact)**, the odd-one-out is always a **(+ Inside, - Contact)** image. This dataset is **Test Other Subroutine**, with respect to the subnetwork that implements **(+/- Inside)**.

**Methods:** Our models consist of a backbone followed by a 2-layer MLP[6], which produces embeddings of each of the four images in an example. Following Zerroug et al. (2022) we compute the dot product between each of the four embeddings to produce a pairwise similarity metric. The least similar embedding is predicted to be the "odd-one-out". We use cross-entropy loss over the four images. During mask training, we use $L_0$ regularization to encourage sparsity. We investigate 3 backbone architectures: Resnet50 [7] (He et al., 2016), Wide Resnet50 (Zagoruyko & Komodakis, 2016), and ViT (Dosovitskiy et al., 2020). We perform a hyperparameter search over batch size and learning rate to find settings that allow each model to achieve near-perfect performance. We then train 3 models with different random seeds in order to probe for structural compositionality.[8]

After training our base models, $M_C$, we perform a hyperparameter search over continuous sparsification parameters for each subroutine (See Appendix C). One hyperparameter to note is the mask configuration: the layer of the network in which to start masking. After finding the best continuous sparsification parameters, we run the algorithm three times per model, per subroutine, and evaluate on

---

[6]Hidden Size: 2048, Output Size: 128

[7]We replace all BatchNorm layers with InstanceNorm layers. BatchNorm statistics learned during training the base model do not apply to the subnetworks, and because the batch statistics vary across the different data partitions that we evaluate on.

[8]All models are trained using the Adam optimizer (Kingma & Ba, 2014) with early stopping for a maximum of 100 epochs (patience set to 75 epochs). We evaluate using a held-out validation set after every epoch and take the model that minimizes loss on the validation set. We train without dropout, as dropout increases a model's robustness to ablating subnetworks. We train without weight decay, as we will apply $L_0$ regularization during mask training.

**Test Target Subroutine** and **Test Other Subroutine**. Finally, for each subnetwork, $Sub_i$, we create $M_{ablate_i} = M_C - Sub_i$ and evaluate it on **Test Target Subroutine** and **Test Other Subroutine**[9].

## 6   Language Experiments

**Tasks:**   We use a subset of the data introduced in Marvin & Linzen (2019) to construct odd-one-out tasks for language data. Analogous to the vision domain, odd-one-out tasks consist of four sentences, three of which follow a rule and one of which does not. We construct rules based on two forms of syntactic agreement: Subject-Verb Agreement and Reflexive Anaphora agreement[10]. In both cases, the agreement takes the form of long-distance coordination of the syntactic number of two words in a sentence. First, consider the subject-verb agreement, the phenomenon that renders *the house near the fields is on fire* grammatical, and *the house near the fields are on fire* not grammatical.

Accordingly, we define the following sentence types for Subject-Verb agreement: **({Singular/Plural} Subject, {Singular/Plural} Verb)**. Because both **(Singular Subject, Singular Verb)** and **(Plural Subject, Plural Verb)** result in a grammatical sentence, we partition the Subject-Verb Agreement dataset into two subsets, one that targets singular sentences and one that targets plural sentences[11]. For the Singular Subject-Verb Agreement dataset, base models are trained on a compositional rule that defines **(Singular Subject, Singular Verb)** sentences to be rule-following, and **(Plural Subject, Singular Verb)** and **(Singular Subject, Plural Verb)** sentences to be the odd-one-out. Thus, an odd-one-out example might look like: *the picture by the ministers interest people.* All other tasks are constructed analogously to those used in the vision experiments (See Section 5). The Reflexive Anaphora dataset is constructed simlarly (See Appendix F).

**Methods:**   The language experiments proceed analogously to the vision experiments. The only difference in the procedure is that we take the representation of the [CLS] token to be the embedding of the sentence and omit the MLP. We study one architecture, BERT-Small (Bhargava et al., 2021; Turc et al., 2019), which is a BERT architecture with 4 hidden layers (Devlin et al., 2018).

## 7   Results

Most base models perform near perfectly, with the exception of ViT, which failed to achieve >90% performance on any of the tasks with any configuration of hyperparameters[12]. Thus, we exclude ViT from all subsequent analyses. See Appendix D for these results. If the base models exhibit structural compositionality, we expect subnetworks to achieve greater accuracy[13] on **Test Target Subroutine** than on **Test Other Subroutine** (difference in accuracies $> 0$). After ablating subnetworks, we expect the ablated model to achieve greater accuracy on **Test Other** than **Test Target** (difference in accuracies $< 0$). Across the board, we see the expected pattern. Subnetwork and ablated accuracy differences for Resnet50 and BERT are visualized in Figure 4 (Subnetwork in Blue, Ablated Models in Red). See Appendix A for Wide Resnet50 results, which largely reproduce the results using Resnet50.

For some architecture/task combinations, the pattern of ablated model results is statistically significantly in favor of structural compositionality. See Figure 4 (C, D), where all base models seem to implement both subroutines in a modular fashion. We analyze the layerwise overlap between subnetworks found within one of these models in Appendix K. This analysis shows that there is relatively high overlap between subnetworks for the same subroutine, and low overlap between subroutines. Other results are mixed, such as those found in Resnet50 models trained on **Number-Contact**. Here,

---

[9]We used NVIDIA GeForce RTX 3090 GPUs for all experiments. Every experiment can be run on a single GPU, in approximately 1 GPU-hour. After performing a hyperparameter search, our main results took approximately 300 GPU-hours.

[10]Note that we are interested only in discovering some evidence of modularity within the model rather than looking for some more profound syntactic phenomenon.

[11]See Appendix F for more details

[12]See Table 2 in Appendix A for each base model's performance on the relevant compositional task.

[13]All accuracy values are clamped to the range [0.25, 1.0] before differences are computed. 0.25 is chance accuracy. Constraining values to this range prevents false trends from arising in the difference data due to models performing below chance.

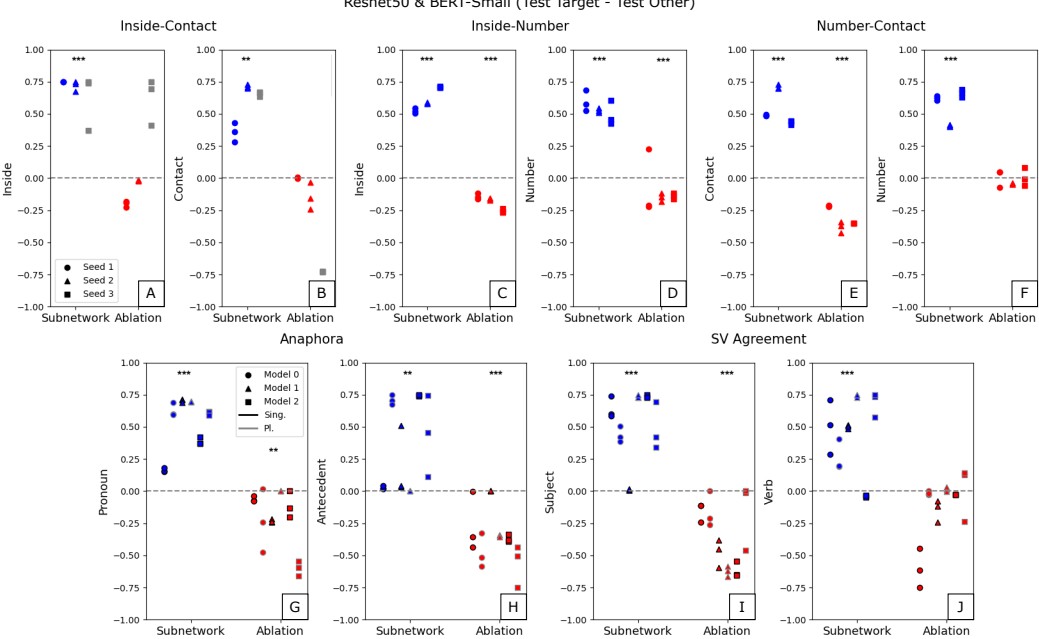

Figure 4: Results from Subnetwork and Ablation studies. For each compositional task, we learn binary masks that result in subnetworks for each subroutine. Resnet50 results in the top row, BERT-Small results in the bottom row. Gray markers indicate that the corresponding base model did not achieve > 90% accuracy on the compositional task. **(Blue)** The difference between subnetwork performance on **Test Target Subroutine** and **Test Other Subroutine**. If a model exhibits structural compositionality, we expect that a subnetwork will achieve greater performance on the subroutine that it was trained to implement, resulting in values > 0. **(Red)** After ablating the subnetwork, we evaluate on the same datasets and plot the difference again. We expect that the ablated model will achieve lower performance on the subroutine that the (ablated) subnetwork was trained to implement and higher performance on the other subroutine dataset, resulting in values < 0. Across the board, we find that our results are largely significantly different from 0, despite the small number of samples. ** indicates significance at p = .01, *** indicates significance at p = .001 See Appendix A for details of this statistical analysis.

we see strong evidence of structural compositionality in Figure 4 (E), but little evidence for it in Figure 4 (F). In this case, it appears that the network is implementing the **(+/- Contact)** subroutine in a small, modular subnetwork, whereas the **(+/- Number)** subroutine is implemented more diffusely. We perform control experiments using randomly initialized models in Appendix I, which show that the pattern of results in (A), (B) and (F) are *not* significantly different from a random model, while all other results *are* significantly different.

## 8 Effect of Pretraining on Structural Compositionality

We compare structural compositionality in models trained from scratch to those that were initialized with pretrained weights. For Resnet50, we pretrain a model on our data using SimCLR (See Appendix G for details).

For BERT-Small, we use the pretrained weights provided by Turc et al. (2019). We rerun the same procedure described in Sections 5 and 6. See Appendix A for each base model's performance. Figure 5 contains the results of the language experiments. Across all language tasks, the ablation results indicate that models initialized with pretrained weights more reliably produce modular subnetworks than randomly initialized models. Results on vision tasks are found in Appendix A and do not suggest any benefit of pretraining.

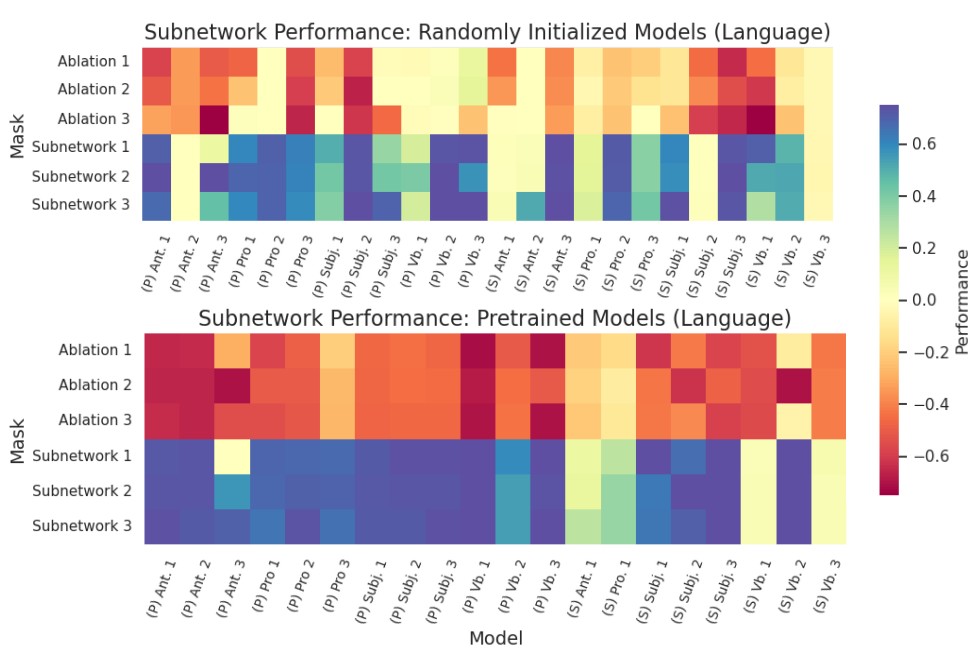

Figure 5: Performance differences between **Test Target Subroutine** and **Test Other Subroutine** for both models trained from scratch and pretrained models. Across the board, we see that pretraining produces more modular subnetworks (i.e., reveal a greater disparity in performance between datasets). Pretraining also appears to make our subnetwork-discovery algorithm more robust to random seeds.

## 9 Related Work

This work casts a new lens on the study of compositionality in neural networks. Most prior work has focused on compositional generalization of standard neural models (Yu & Ettinger, 2020; Kim & Linzen, 2020; Kim et al., 2022; Dankers et al., 2022), though some has attempted to induce an inductive bias toward compositional generalization from data (Lake, 2019; Qiu et al., 2021; Zhu et al., 2021). Recent efforts have attempted to attribute causality to specific components of neural networks' internal representations (Ravfogel et al., 2020; Bau et al., 2019; Wu et al., 2022; Tucker et al., 2021; Lovering & Pavlick, 2022; Elazar et al., 2021; Cao et al., 2021). In contrast to these earlier studies, our method does not require any assumptions about where in the network the subroutine is implemented and does not rely on auxiliary classifiers, which can confound the causal interpretation. Finally, Dziri et al. (2023) performs extensive behavioral studies characterizing the ability of autoregressive language models to solve compositional tasks, and finds them lacking. In contrast, our work studies the structure of internal representations and sets aside problems that might be specific to autoregressive training objectives.

More directly related to the present study is the burgeoning field of *mechanistic interpretability*, which aims to reverse engineer neural networks in order to better understand how they function (Olah, 2022; Cammarata et al., 2020; Black et al., 2022; Henighan et al., 2023; Ganguli et al., 2021; Merrill et al., 2023). Notably, Chughtai et al. (2023) recovers universal mechanisms for performing group-theoretic compositions. Though group-theoretic composition is different from the compositionality discussed in the present article, this work sheds light on generic strategies that models may use to solve tasks that require symbolically combining multiple input features.

Some recent work has attempted to characterize modularity within particular neural networks (Hod et al., 2022). Csordás et al. (2021) also analyzes modularity within neural networks using learned binary masks. Their study finds evidence of modular subnetworks within a multitask network: Within a network trained to perform both addition and multiplication, different subnetworks arise for each operation. Csordás et al. (2021) also investigates whether the subnetworks are reused in a variety of contexts, and find that they are not. In particular, they demonstrate that subnetworks that solve particular partitions of compositional datasets (SCAN (Lake & Baroni, 2018) and the Mathematics

Dataset (Saxton et al., 2018)), oftentimes do not generalize to other partitions. From this, they conclude that neural networks do not flexibly combine subroutines in a manner that would enable full compositional generalization. However, their work did not attempt to uncover subnetworks that implement specific compositional subroutines within these compositional tasks. For example, they did not attempt to find a subnetwork that implements a general "repeat" operation for SCAN, transforming "jump twice" into "JUMP JUMP". Our work finds such compositional subroutines in language and vision tasks, and localizes them into modular subnetworks. This finding extends Csordás et al. (2021)'s result on a simple multitask setting to more complex compositional vision and language settings, and probes for subroutines that represent intermediate subroutines in a compositional task (i.e. "inside" is a subroutine when computing "Inside-Contact").

## 10   Discussion

Across a variety of architectures, tasks, and training regimens, we demonstrated that models often exhibit structural compositionality. Without any explicit encouragement to do so, neural networks appear to decompose tasks into subroutines and implement solutions to (at least some of) these subroutines in modular subnetworks. Furthermore, we demonstrate that self-supervised pretraining can lead to more consistent structural compositionality, at least in the domain of language. These results bear on the longstanding debate over the need for explicit symbolic mechanisms in AI systems. Much work is focusing on integrating symbolic and neural systems (Ellis et al., 2023; Nye et al., 2020). However, our results suggest that some simple pseudo-symbolic computations might be learned directly from data using standard gradient-based optimization techniques.

We view our approach as a tool for understanding when and how compositionality arises in neural networks, and plan to further investigate the conditions that encourage structural compositionality. One promising direction would be to investigate the relationship between structural compositionality and recent theoretical work on compositionality and sparse neural networks (Mhaskar & Poggio, 2016; Poggio, 2022). Specifically, this theoretical work suggests that neural networks optimized to solve compositional tasks naturally implement sparse solutions. This may serve as a starting point for developing a formal theory of structural compositionality in neural networks. Another direction might be to investigate the structural compositionality of networks trained using iterated learning procedures (Ren et al., 2019; Vani et al., 2020). Iterated learning simulates the cultural evolution of language by jointly training two communicating agents (Kirby et al., 2008). Prior work has demonstrated that iterated learning paradigms give rise to simple compositional languages. Quantifying the relationship between structural compositionality within the agents and the compositionality of the language that they produce would be an exciting avenue for understanding the relationship between representation and behavior.

One limit of our technical approach is that one must specify which subroutines to look for in advance. Future work might address this by discovering functional subnetworks using unsupervised methods. Additionally, our approach requires us to use causal ablations and control models to properly interpret our results. Future work might try to uncover subnetworks that are necessarily causally implicated in model behavior. Finally, future work must clarify the relationship between structural compositionality and compositional generalization.

## Acknowledgments and Disclosure of Funding

The authors thank the members of the Language Understanding and Representation Lab and Serre Lab for their valuable feedback on this project and Rachel Goepner for proofreading the manuscript. This project was supported by ONR grant #N00014-19-1-2029. The computing hardware was supported in part by NIH Office of the Director grant #S10OD025181 via the Center for Computation and Visualization (CCV) at Brown University.

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

| COMP. TASK | MODEL | SR | INTERCEPT Z | LINEAR HYPOTHESIS $\chi^2$ |
|---|---|---|---|---|
| IN. CONT. | RN50 | INSIDE | 50.13*** | 1.42 |
| IN. CONT. | RN50 | CONTACT | 3.00** | 0.94 |
| IN. CONT. | WRN50 | INSIDE | 6.66*** | 0.87 |
| IN. CONT. | WRN50 | CONTACT | 72.44*** | 4.49* |
| IN. NUM. | RN50 | INSIDE | 11.13*** | 29.20*** |
| IN. NUM. | RN50 | NUMBER | 18.35*** | 21.73*** |
| IN. NUM. | WRN50 | INSIDE | 14.86*** | 2.95. |
| IN. NUM. | WRN50 | NUMBER | 5.21*** | 2.50 |
| CONT. NUM. | RN50 | CONTACT | 6.45*** | 40.26*** |
| CONT. NUM. | RN50 | NUMBER | 7.12*** | 0.42 |
| CONT. NUM. | WRN50 | CONTACT | 6.19*** | 296.96*** |
| CONT. NUM. | WRN50 | NUMBER | 9.388*** | 1.32 |
| SV AGR | BERT | SUBJ. | 4.53*** | 14.72*** |
| SV AGR | BERT | VERB | 3.73*** | 1.70 |
| ANAPHORA | BERT | PRONOUN | 6.06*** | 5.60* |
| ANAPHORA | BERT | ANTECEDENT | 2.60** | 17.83*** |

Table 1: Statistics from one factor GLM with robust clustered standard errors. . indicates significance at p = .1, * indicates significance at p = .05, ** indicates significance at p = .01, *** indicates significance at p = .001

# A   Full Results

In this section, we provide the following results:

1. Base Model Performance on Compositional Tasks: Table 2

2. Pretrained + Finetuned Model Performance on Compositional Tasks: Table 3

3. Wide Resnet50 Subnetwork Results: Figure 6

4. Vision Pretraining vs. Random Initialization Heatmap: Figure 7

5. Absolute Accuracy for every subnetwork and ablated model on each task, for each model: Figures 8-12.

See Table 2 for the performance of all base models on each compositional task. See Table 3 for the performance of all pretrained base models on each compositional task. See Figure 6 for subnetwork and ablation results on Wide Resnet50. See Figure 7 for Vision Model pretraining results.

## A.1   Statistical Analysis of Main Results

In order to assess the significance of our main results, we fit a generalized linear model (GLM) with robust clustered standard errors for each combination of model architecture, compositional task, and subroutine. This GLM includes a dummy variable indicating whether the results are from a subnetwork or an ablated model, and it clusters observations by base model. For language experiments, we collapse across singular and plural instances of the same subroutine. The intercept term in this model assesses whether the performance of the discovered subnetworks are significantly different from 0. From this model, we can also perform a linear hypothesis test to assess whether ablated model performance is significantly different from 0. Table 1 provides the relevant statistics. Across the board, we see that there is always a significant difference between subnetwork performance and 0, and often a significant difference between ablated model performance and 0, even with a small sample size.

| VISION | CONT.-INSIDE | CONT.-NUMBER | INSIDE-NUMBER | |
|---|---|---|---|---|
| RN50-1 | 100% | 99.4% | 99.8% | |
| RN50-2 | 100% | 99.4% | 99.8% | |
| RN50-3 | 75.9% | 99.7% | 99.9% | |
| WRN50-1 | 99.9% | 99.6% | 99.7% | |
| WRN50-2 | 99.8% | 99.8% | 99.6% | |
| WRN50-3 | 99.9% | 99.4% | 99.8% | |
| LANGUAGE | SV SING. | SV PLUR. | ANAPH. SING. | ANAPH. PLUR. |
| BERT-SM-1 | 99.7% | 100% | 100% | 100% |
| BERT-SM-2 | 100% | 100% | 100% | 100% |
| BERT-SM-3 | 100% | 100% | 100% | 100% |

Table 2: Test classification accuracy for each base model for each task. Every entry corresponds to a unique model.

| VISION | CONT.-INSIDE | CONT.-NUMBER | INSIDE-NUMBER | |
|---|---|---|---|---|
| RN50-SC-1 | 100% | 99.7% | 100% | |
| RN50-SC-2 | 100% | 99.6% | 99.8% | |
| RN50-SC-3 | 100% | 99.5% | 99.9% | |
| LANGUAGE | SV SING. | SV PLUR. | ANAPH. SING. | ANAPH. PLUR. |
| BERT-LM-1 | 100% | 100% | 100% | 100% |
| BERT-LM-2 | 100% | 100% | 65.5% | 100% |
| BERT-LM-3 | 100% | 100% | 63.5% | 100% |

Table 3: Test classification accuracy for each pretrained base model for each task. Every entry corresponds to a unique model.

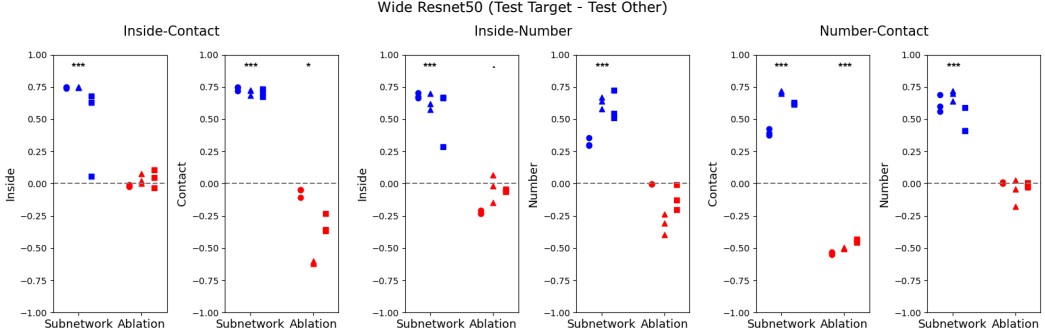

Figure 6: Wide Resnet50 Subnetwork and Ablation Results. Broadly, they mimic those found in Figure 4. . indicates significance at p = .1, * indicates significance at p = .05, ** indicates significance at p = .01, *** indicates significance at p = .001.

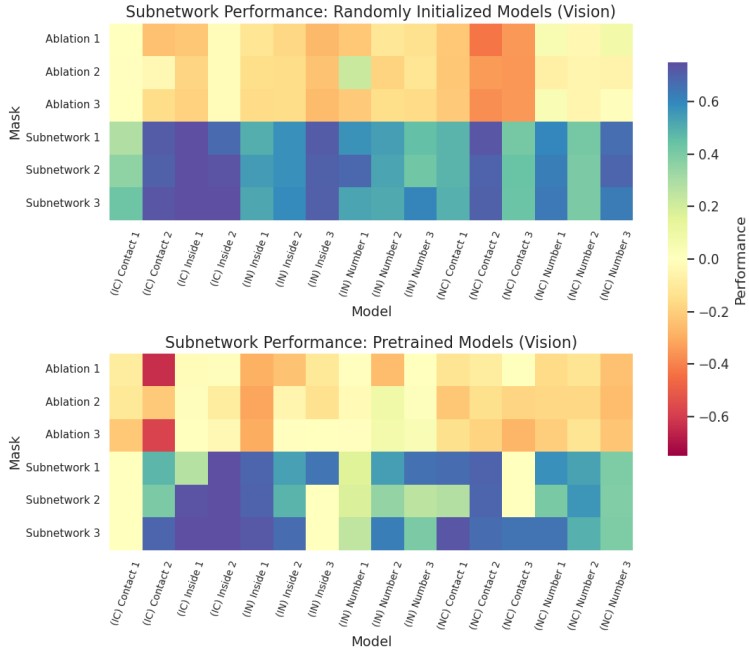

Figure 7: Vision Model Pretraining vs. Random Initialization. We observe no obvious trend differentiating the two conditions.

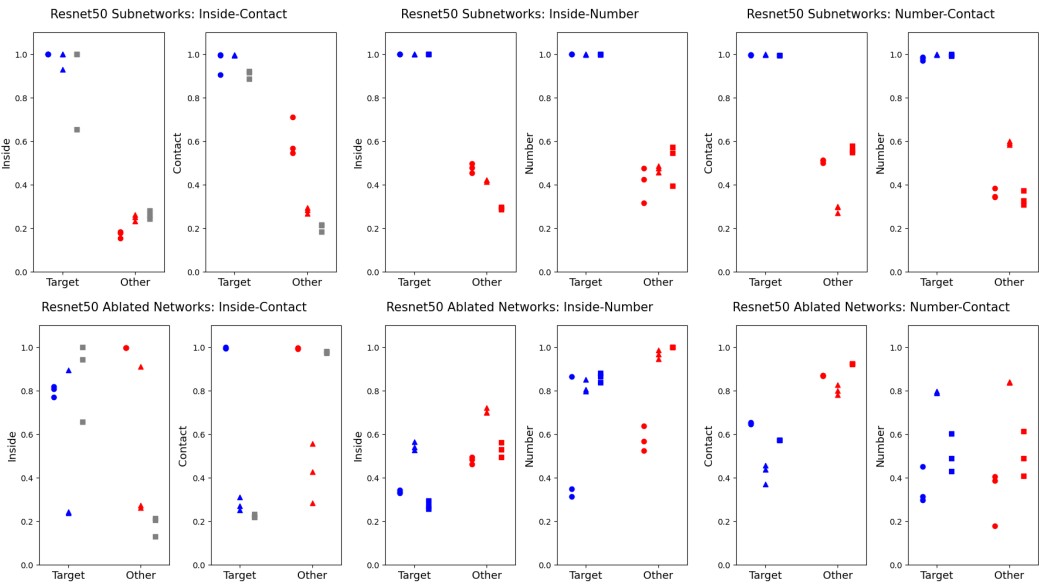

Figure 8: Resnet50 absolute performance across all conditions.

# B  Continuous Sparsification: Extended Discussion

Continuous sparsification attempts to optimize a binary mask that minmizes the following loss function:

$$\min_{m_i \in \{0,1\}^d} L_{SR_i}(M_C(\cdot; w \odot m_i)) + \lambda||m_i||_1 \tag{1}$$

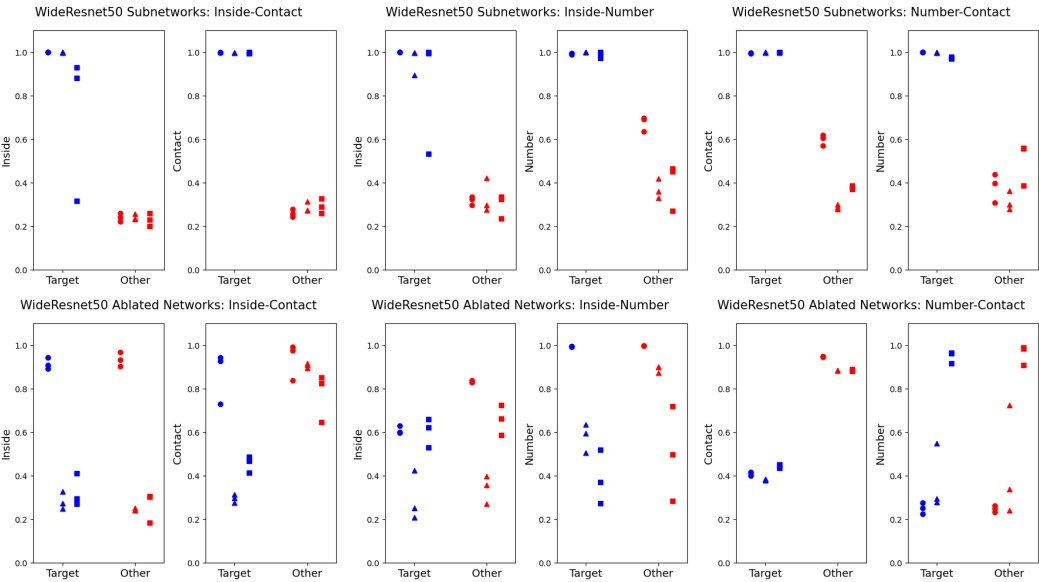

Figure 9: Wide Resnet50 absolute performance across all conditions

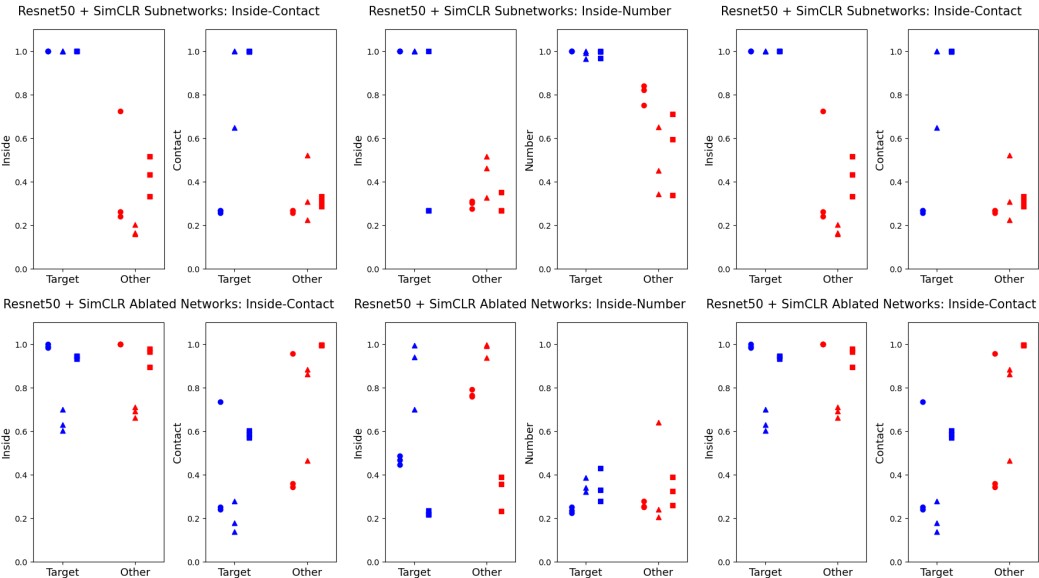

Figure 10: Resnet50 + SimCLR absolute performance across all conditions.

The first term describes the standard loss function given by an odd-one-out task where the rule is defined by $SR_i$. The second term corresponds to the $L_0$ penalty, which encourages entries in the binary mask to be 0. However, optimizing such a binary mask is intractable, given the combinatorial nature of a discrete binary mask over a large parameter space. Instead, continuous sparsification reparameterizes the loss function by introducing another variable, $s \in \mathbb{R}^d$:

$$\min_{s_i \in \mathbb{R}^d} L_{SR_i}\big(M_C(\cdot; w \odot \sigma(\beta \cdot s_i)) + \lambda ||\sigma(\beta \cdot s_i)||_1 \tag{2}$$

In Equation 2, $\sigma$ is the sigmoid function, applied elementwise, and $\beta$ is a temperature parameter. During training $\beta$ is increased after each epoch according to an exponential schedule to a large value $\beta_{max}$. Note that, as $\beta \to \infty$, $\sigma(\beta \cdot s_i) \to H(s_i)$, where $H(s_i)$ is the *heaviside function*.

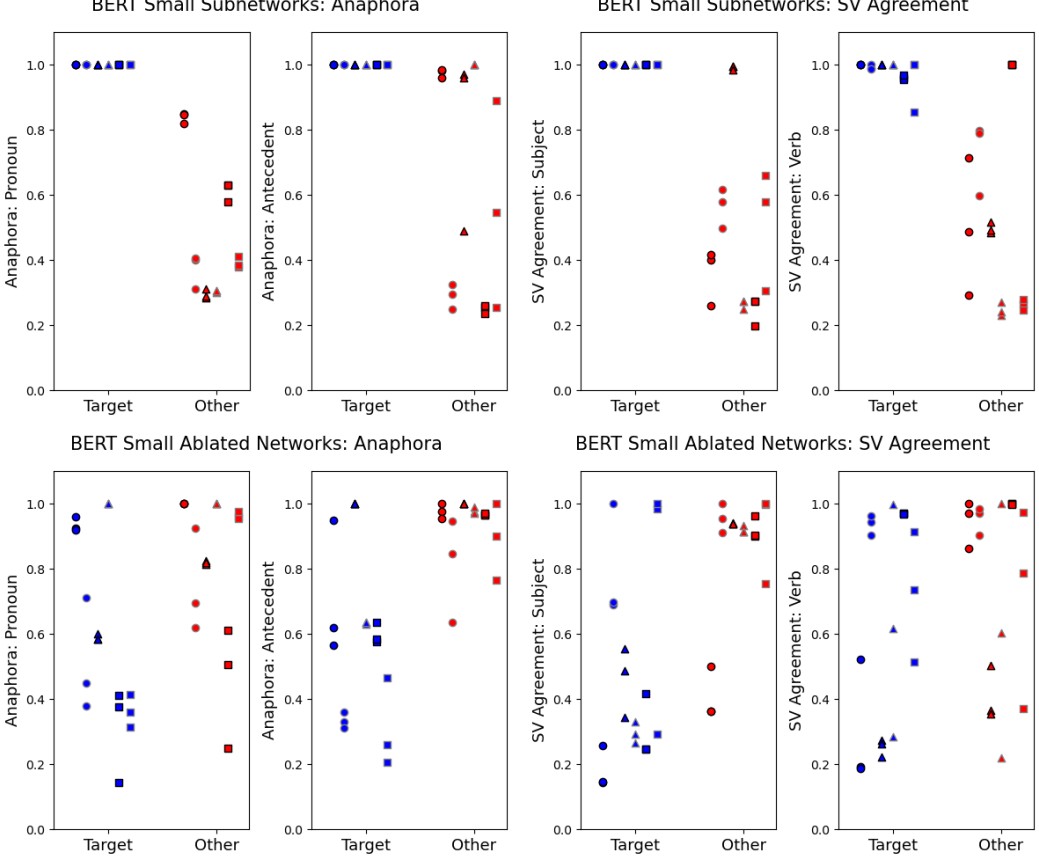

Figure 11: BERT-Small absolute performance across all conditions

$$H(s) = \left\{ \begin{array}{l} 0, s < 0 \\ 1, s > 0 \end{array} \right\} \tag{3}$$

Thus, during training, we interpolate between a soft mask ($\sigma$) and a discrete mask ($H$). During inference, we simply substitute $\sigma(\beta_{max} \cdot s_i)$ for $H(s_i)$. Notably, we apply continuous sparsification to a frozen model in an attempt to reveal the internal structure of this model, whereas the original work introduced continuous sparsification in the context of model pruning, and jointly trained $w$ and $s$.

Following Savarese et al. (2020), we fix $\beta_{max} = 200$, $\lambda = 10^{-8}$, and train for 90 epochs. We train the mask parameters using the Adam optimizer with a batch size of 64 and search over learning rates.

## C   Mask Hyperparameter Search Details

We search over learning rates {.01, .0001}, mask parameter initializations {0.1, 0.05, 0.0, -0.05}, and mask configurations. For Resnet models, we search over mask configurations by starting masking at different stages. We try either (1) masking the whole network, (2) beginning masking at the third (of four) stages), and (3) beginning masking at the fourth stage. For transformer models, we search over mask configurations based on layers. We try either (1) masking the whole networks, (2) beginning masking at the third (of four) layers, (3) beginning masking at the fourth layer.

We perform this search independently for each trained model and each subroutine. The best hyperparameter configuration is determined based on the following criteria: The subnetwork must achieve at least 90% accuracy on the task it was trained on. This is to ensure that mask optimization succeeded. Then, it was scored on its degree of structural compositionality using the validation sets

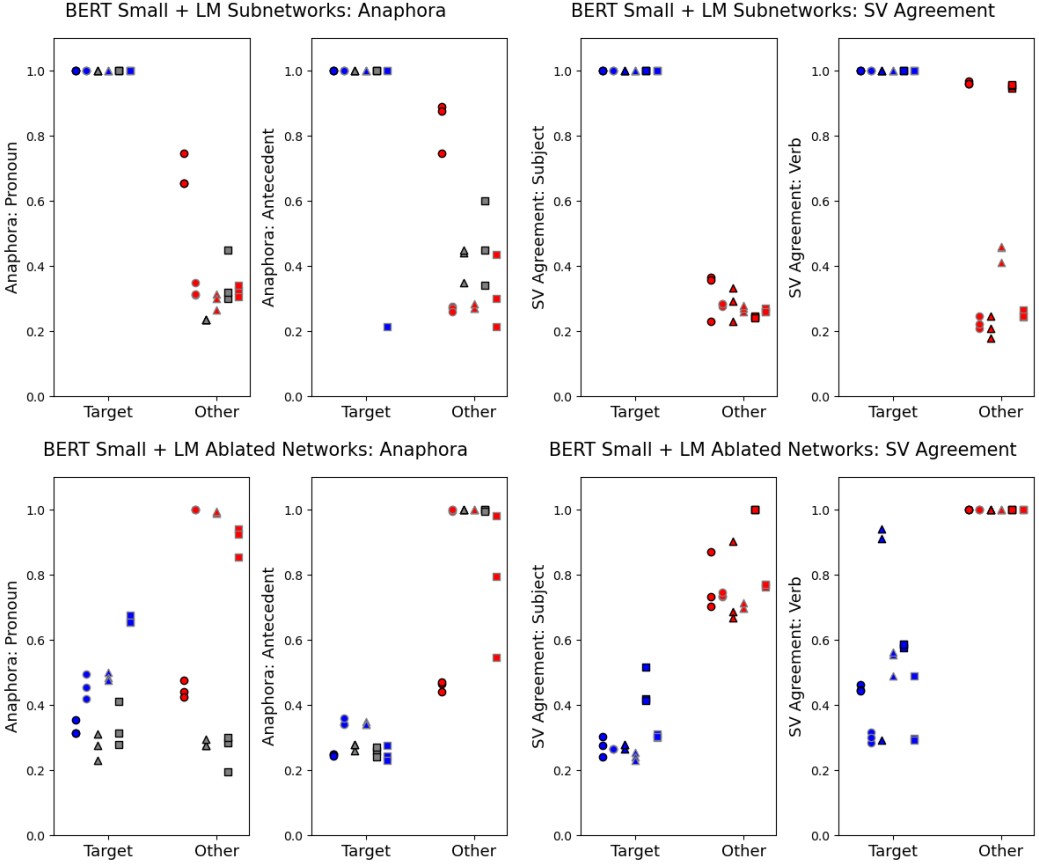

Figure 12: BERT-Small + LM absolute performance across all conditions.

of **Test Target Subroutine** and **Test Other Subroutine** If a subnetwork is trained to implement $SR_1$, then its compostionality score is calculated using $M_{ablate_1} = M_C - Sub_1$. The score is simply the difference in accuracy that $M_{ablate_1}$ achieves on **Test Other Subroutine** (which the ablated model should perform well on) and **Test Target Subroutine** (which the ablated model should fail on). All accuracies are clamped in the range [.25, 1], as .25 is chance accuracy. The hyperparameters that maximize this score are returned.

Note that this process is fairly computationally expensive, as it requires training many separate masks. We used NVIDIA GeForce RTX 3090 GPUs for all experiments. Every experiment can be run on a single GPU, in approximately 1 GPU-hour. The entire hyperparameter search takes approximately 2448 GPU-hours. This number was computed as follows: 1 GPU-hour * 3 model seeds * 2 learning rates * 4 initializations * 3 mask configurations * (6 Resnet50 subroutines + 6 pretrained Resnet50 subroutines + 6 Wide Resnet50 subroutines + 8 BERT subroutines + 8 pretrained BERT subroutines). Each mask has a parameter count comparable to its base model. Future work could improve upon the methodology presented here by reducing the number of hyperparameters that one must search over.

# D  ViT Hyperparameter Search Results

See Table 4 for the results of our hyperparameter search on ViT models. We tried several batch sizes and learning rates on both a 6 and 12 layer ViT, all with a 2 layer MLP head. The MLP had a hidden layer of dimensionality 2048, and an output dimensionality of 128, similar to the Resnet50 and Wide Resnet50. Note that all models fall short of solving any of the tasks.

| # Layers | Batch Size | Learning Rate | Cont.-Inside | Cont.-Number | Inside-Number |
|---|---|---|---|---|---|
| 6 | 32 | 0.01 | 31% | 27% | 28% |
| 6 | 64 | 0.01 | 28% | 29% | 27% |
| 6 | 32 | 0.001 | 29% | 27% | 28% |
| 6 | 64 | 0.001 | 32% | 27% | 26% |
| 6 | 32 | 0.0001 | 25% | 32% | 30% |
| 6 | 64 | 0.0001 | 31% | 49% | 32% |
| 6 | 32 | 0.00001 | 42% | 85% | 47% |
| 6 | 64 | 0.00001 | 46% | 83% | 54% |
| 12 | 32 | 0.01 | 36% | 25% | 28% |
| 12 | 64 | 0.01 | 27% | 26% | 25% |
| 12 | 32 | 0.001 | 39% | 31% | 27% |
| 12 | 64 | 0.001 | 37% | 25% | 22% |
| 12 | 32 | 0.0001 | 41% | 36% | 33% |
| 12 | 64 | 0.0001 | 31% | 28% | 31% |
| 12 | 32 | 0.00001 | 40% | 86% | 49% |
| 12 | 64 | 0.00001 | 42% | 84% | 51% |

Table 4: Results of ViT hyperparameter search. All accuracies are rounded to the nearest % and are computed on the validation set for the relevant dataset.

## E   Vision Stimuli

In this section, we provide examples from all vision datasets that we use in this work. First, we describe the **+/- Number** subroutine. This subroutine operates as follows: for each training/test example, let $N$ be an integer. All image types that exhibit **(+ Number)** will contain $N$ shapes, whereas image types that exhibit **(- Number)** will contain $M$ shapes, $M \neq N$. For a description of the other subroutines, refer back to Section 5.

## F   Language Data Details

As noted in the main text, our language data is generated using the templates provided by Marvin & Linzen (2019). For the subject-verb agreement datasets, we omit templates that position the noun of interest inside either a sentential complement or an object relative clause. Thus, all of our nouns of interest are the subject of the full sentence. This is done in order to render the **(Singular/Plural Subject)** subroutine unambiguous across different sentence templates. We do the same for the Reflexive Anaphora datasets, removing the template that positions the antecedent inside a sentential complement.

These exclusions mean that the nouns of interest are always the second word of the sentence. This makes the **(Singular/Plural Subject)** subroutine amenable to a simple heuristic: check the syntactic number of the second word in the sentence, rather than first needing to identify the subject of a sentence. However, we are unconcerned about this heuristic: the present work makes no claims about how a neural network implements any *particular* subroutine, instead caring about how *several subroutines* are organized in the network's weights (i.e. are they represented compositionally, or in an entangled fashion?).

Specifically, the subroutines we examine are those that compute the syntactic number of specific words in a sentence (either subject and verb, or antecedent and pronoun). Our goal is to find subnetworks that implement these subroutines. Consider the case of subject verb agreement. If we were to partition our data precisely analogously to the vision datasets, we would arrive at a compositional dataset where rule-following data points exhibit, say, **(Singular Subject, Singular Verb)**, and rule breaking examples might exhibit any of **(Plural Subject, Singular Verb)**, **(Singular Subject, Plural Verb)**, or **(Plural Subject, Plural Verb)**. However, one might expect that a pretrained network would implement syntactic number subroutines in service of another salient computation: discerning whether a sentence is grammatical or not. In this case, a pretrained model would need to

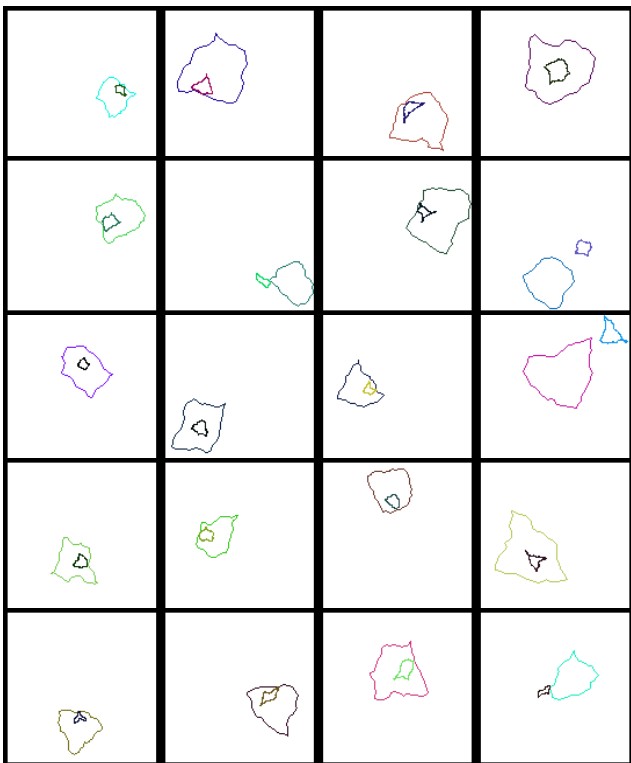

Figure 13: Examples from tasks defined over the **Inside-Contact** compositional rule. From top to bottom, we see one example from each of the following tasks: (1) The task used to train base models (2) The task used to train a **+/- Contact** subnetwork (3) The task used to train a **+/- Inside** subnetwork (4) The evaluation task used to probe for **+/- Contact** (5) The evaluation task used to probe for **+/- Inside**.

unlearn this grammaticality computation, forcing two grammatical sentences apart in its embedding space. In order to avoid this potential complication, we split up our datasets into singular and plural partitions, such that only rule-following examples are grammatical (and all rule-breaking examples are ungrammatical) in each compositional dataset and subroutine test set.

Note that these datasets are smaller than those used for the vision experiments. Using the Marvin & Linzen (2019) templates and their provided vocabulary, and discarding the templates noted above, we arrive at the following dataset statistics (which are identical for the singular and plural instances of each dataset). For each, we provide on singular example and one plural example. The odd one out is always the fourth sentence.

- **Subject-Verb Agreement: Compositional Dataset**: 9500 (Train), 500 (Validation), 1000 (Test)

  **Singular**

  1. the farmer near the parent is old
  2. the surgeon that the architects hate laughs
  3. the novel that the dancer likes is new
  4. the senator to the side of the parents are young

  **Plural**

  1. the farmers the taxi drivers love are short
  2. the songs the dancers admire are unpopular
  3. the surgeons that admire the executives are young
  4. the officers that love the assistant is short

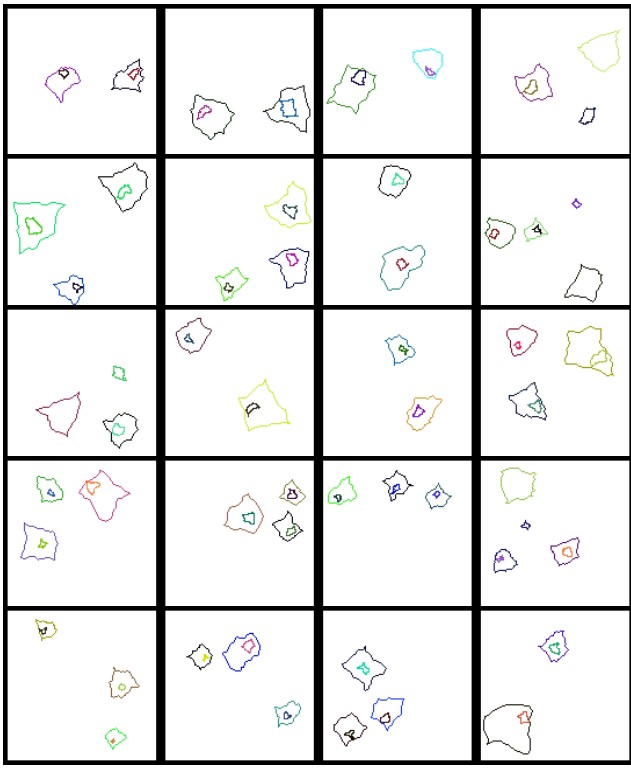

Figure 14: Examples from tasks defined over the **Inside-Number** compositional rule. From top to bottom, we see one example from each of the following tasks: (1) The task used to train base models (2) The task used to train a **+/- Inside** subnetwork (3) The task used to train a **+/- Number** subnetwork (4) The evaluation task used to probe for **+/- Inside** (5) The evaluation task used to probe for **+/- Number**.

- **Subject-Verb Agreement: (Singular/Plural Subject) Dataset**: 9500 (Train), 500 (Validation), 1000 (Test)

  **Singular**

    1. the farmer that the taxi driver hates smile
    2. the consultant the guards hate is young
    3. the poem that the assistant likes brings joy to people
    4. the customers that the chefs like is tall

  **Plural**

    1. the novels the guard hates are good
    2. the teachers across from the parent is young
    3. the shows that the taxi driver likes are unpopular
    4. the manager across from the parent smile

- **Subject-Verb Agreement: (Singular/Plural Verb) Dataset**: 9500 (Train), 500 (Validation), 1000 (Test)

  **Singular**

    1. the game the executives admire is unpopular
    2. the surgeon to the side of the taxi drivers smiles
    3. the consultants the dancer likes swims
    4. the painting that the chefs love are unpopular

  **Plural**

    1. the customer the assistant loves swim

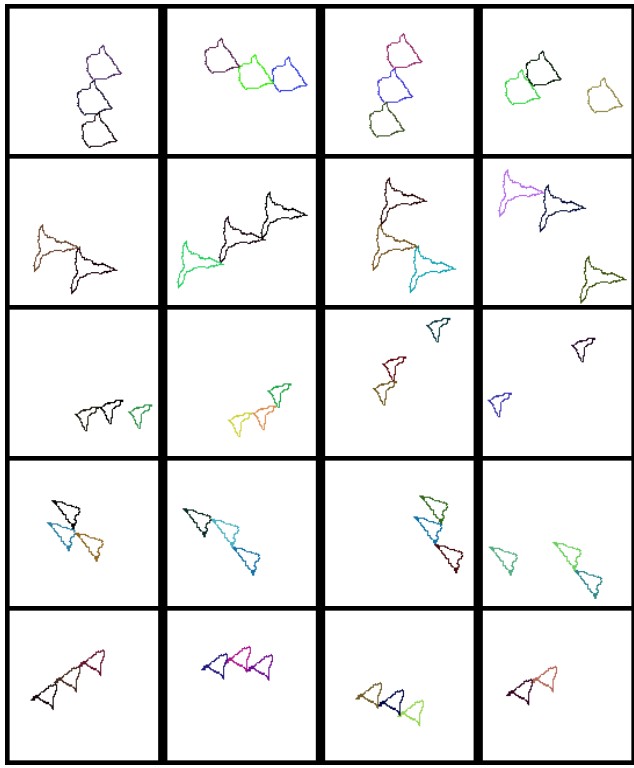

Figure 15: Examples from tasks defined over the **Number-Contact** compositional rule. From top to bottom, we see one example from each of the following tasks: (1) The task used to train base models (2) The task used to train a **+/- Contact** subnetwork (3) The task used to train a **+/- Number** subnetwork (4) The evaluation task used to probe for **+/- Contact** (5) The evaluation task used to probe for **+/- Number**.

2. the surgeons that the executive likes are short
3. the authors that love the chef swim
4. the officers that like the assistant swims

- **Subject-Verb Agreement: Test (Singular/Plural Subject) Dataset**: 300 (Validation), 300 (Test)

  **Singular**

  1. the teacher to the side of the taxi driver swims
  2. the farmer that the chef likes is young
  3. the novel the ministers admire is bad
  4. the customers in front of the dancers is old

  **Plural**

  1. the pictures by the skater interest people
  2. the movies the skater admires are bad
  3. the pilots in front of the taxi driver are tall
  4. the consultant that the dancer likes are old

- **Subject-Verb Agreement: Test (Singular/Plural Verb) Dataset**: 300 (Validation), 300 (Test)

  **Singular**

  1. the senator the taxi drivers admire is young
  2. the pilot to the side of the dancer smiles
  3. the farmer the assistant admires is young

4. the pilot that loves the minister are tall

**Plural**

1. the poems that the chefs hate are bad
2. the surgeons in front of the dancer laugh
3. the surgeons near the taxi drivers smile
4. the farmers behind the architects is short

- **Reflexive Anaphora: Compositional Dataset**: 2500 (Train), 200 (Validation), 200 (Test)
**Singular**

1. the consultant that the chef loves disguised himself
2. the manager that the architects hate congratulated herself
3. the pilot that the architects admire hurt herself
4. the surgeon that the executives like congratulated themselves

**Plural**

1. the consultants that the guards love injured themselves
2. the senators that the minister admires embarrassed themselves
3. the officers that the assistant likes embarrassed themselves
4. the teacher that the dancer loves embarrassed themselves

- **Reflexive Anaphora: (Singular/Plural Antecedent) Dataset**: 2500 (Train), 200 (Validation), 200 (Test)
**Singular**

1. the officer that the taxi driver likes doubted herself
2. the author that the architect loves hated himself
3. the manager that the executives love disguised herself
4. the customers that the parent likes disguised himself

**Plural**

1. the authors that the skater hates doubted himself
2. the surgeons that the parents admire hurt themselves
3. the officers that the taxi driver hates injured himself
4. the pilot that the assistant loves hurt themselves

- **Reflexive Anaphora: (Singular/Plural Pronoun) Dataset**: 2500 (Train), 200 (Validation), 200 (Test)
**Singular**

1. the customer that the ministers hate congratulated herself
2. the surgeons that the dancers like embarrassed himself
3. the authors that the taxi driver hates embarrassed himself
4. the author that the architect admires doubted themselves

**Plural**

1. the officer that the skaters admire embarrassed themselves
2. the senator that the guard likes embarrassed themselves
3. the customer that the ministers love doubted themselves
4. the managers that the guard admires injured herself

- **Reflexive Anaphora: Test (Singular/Plural Antecedent) Dataset**: 200 (Validation), 200 (Test)
**Singular**

1. the customer that the skater admires hurt herself
2. the consultant that the executive loves disguised herself
3. the manager that the skaters like embarrassed herself
4. the senators that the guard admires injured himself

**Plural**

1. the senators that the architects like embarrassed themselves
2. the authors that the executives admire disguised themselves
3. the surgeons that the taxi driver admires doubted themselves
4. the officer that the parents like congratulated themselves

- **Reflexive Anaphora: Test (Singular/Plural Pronoun) Dataset**: 200 (Validation), 200 (Test)

**Singular**

1. the pilot that the chefs hate hurt himself
2. the teacher that the taxi drivers love hated herself
3. the senator that the assistant loves embarrassed herself
4. the pilot that the skaters admire embarrassed themselves

**Plural**

1. the authors that the parents admire congratulated themselves
2. the pilots that the chef hates hurt themselves
3. the authors that the parents like congratulated themselves
4. the farmers that the ministers admire injured herself

## G    Vision Pretraining Details

We pretrain a Resnet50 model and MLP using SimCLR, a contrastive self-supervised learning algorithm (Chen et al., 2020). This algorithm generates two views of an image using random data augmentations, then maximizes the agreement between representations of these views using a contrastive loss function. We use a temperature of 0.07 for this loss.

Our data augmentations include horizontal flips, affine transformations, color jitters, rotations, and grayscaling. We train for 100 epochs, using a learning rate of .0005 (which is decayed according to a consine annealing schedule) and a batch size of 256. Images are drawn randomly from the three compositional training sets (**Inside-Contact**, **Number-Contact**, **Inside-Number**). For every rule-following image that is selected, a rule-breaking image from that same dataset is also selected.

We evaluate the Top-5 Accuracy on a held-out validation set after every epoch, and save the weights of the best performing model. Following Chen et al. (2020), we discard the MLP after pretraining, only using the Resnet50 weights to initialize our pretrained models in Section 8.

We adapted the implementation found in Lippe (2022) to implement SimCLR pretraining.

## H    Subnetwork Sparsity Data

In this section, we provide the raw sparsity statistics for each subnetwork trained in this paper. For every subnetwork, we indicate what stage we started masking (0, 3, or 4), provide the number of Act. Param. in the subnetwork (i.e. the number of 1's in the binary mask), and include the total number of parameters that we mask over (i.e. the number of entries in the binary mask), which is determined by the mask stage.

## I    Control Experiment: Random Models

Recent work has demonstrated several surprising properties of masks trained on randomly-intialized networks (Ramanujan et al., 2020; Zhou et al., 2019; Wortsman et al., 2020). One might wonder whether the results demonstrated here could be obtained by training a binary mask over randomly-initialized network. If so, this would pose a serious problem for our interpretation of the data: producing the same results in a randomly-initialized network would decouple the behavior of the discovered subnetworks from the representations learned by the underlying base model.

We carry out this experiment as a control. Specifically, we run the exact same mask training procedure used to generate the results in Section 7, except we use randomly-initialized models rather than

| Task | SR | Model # | Mask # | Stage | Act. Param. | Tot. Param. |
|---|---|---|---|---|---|---|
| Number-Contact | Contact | 1 | 1 | 4 | 8059720 | 19398656 |
| Number-Contact | Contact | 1 | 2 | 4 | 8200381 | 19398656 |
| Number-Contact | Contact | 1 | 3 | 4 | 8199621 | 19398656 |
| Number-Contact | Number | 1 | 1 | 0 | 259611 | 27911360 |
| Number-Contact | Number | 1 | 2 | 0 | 264197 | 27911360 |
| Number-Contact | Number | 1 | 3 | 0 | 438332 | 27911360 |
| Number-Contact | Contact | 2 | 1 | 3 | 996278 | 26476544 |
| Number-Contact | Contact | 2 | 2 | 3 | 1010779 | 26476544 |
| Number-Contact | Contact | 2 | 3 | 3 | 970624 | 26476544 |
| Number-Contact | Number | 2 | 1 | 4 | 1831379 | 19398656 |
| Number-Contact | Number | 2 | 2 | 4 | 1822564 | 19398656 |
| Number-Contact | Number | 2 | 3 | 4 | 1814451 | 19398656 |
| Number-Contact | Contact | 3 | 1 | 4 | 6044544 | 19398656 |
| Number-Contact | Contact | 3 | 2 | 4 | 6111795 | 19398656 |
| Number-Contact | Contact | 3 | 3 | 4 | 6258924 | 19398656 |
| Number-Contact | Number | 3 | 1 | 0 | 225866 | 27911360 |
| Number-Contact | Number | 3 | 2 | 0 | 685279 | 27911360 |
| Number-Contact | Number | 3 | 3 | 0 | 219003 | 27911360 |
| Inside-Contact | Inside | 1 | 1 | 4 | 2418223 | 19398656 |
| Inside-Contact | Inside | 1 | 2 | 4 | 2384556 | 19398656 |
| Inside-Contact | Inside | 1 | 3 | 4 | 2307045 | 19398656 |
| Inside-Contact | Contact | 1 | 1 | 4 | 1029149 | 19398656 |
| Inside-Contact | Contact | 1 | 2 | 4 | 1057758 | 19398656 |
| Inside-Contact | Contact | 1 | 3 | 4 | 938794 | 19398656 |
| Inside-Contact | Inside | 2 | 1 | 3 | 189266 | 26476544 |
| Inside-Contact | Inside | 2 | 2 | 3 | 140762 | 26476544 |
| Inside-Contact | Inside | 2 | 3 | 3 | 184485 | 26476544 |
| Inside-Contact | Contact | 2 | 1 | 3 | 1266369 | 26476544 |
| Inside-Contact | Contact | 2 | 2 | 3 | 1333968 | 26476544 |
| Inside-Contact | Contact | 2 | 3 | 3 | 1078198 | 26476544 |
| Inside-Contact | Inside | 3 | 1 | 3 | 94253 | 26476544 |
| Inside-Contact | Inside | 3 | 2 | 3 | 159807 | 26476544 |
| Inside-Contact | Inside | 3 | 3 | 3 | 231113 | 26476544 |
| Inside-Contact | Contact | 3 | 1 | 3 | 1681559 | 26476544 |
| Inside-Contact | Contact | 3 | 2 | 3 | 1684466 | 26476544 |
| Inside-Contact | Contact | 3 | 3 | 3 | 1682888 | 26476544 |
| Inside-Number | Inside | 1 | 1 | 4 | 7004974 | 19398656 |
| Inside-Number | Inside | 1 | 2 | 4 | 7077554 | 19398656 |
| Inside-Number | Inside | 1 | 3 | 4 | 7077890 | 19398656 |
| Inside-Number | Number | 1 | 1 | 3 | 3190833 | 26476544 |
| Inside-Number | Number | 1 | 2 | 3 | 3291780 | 26476544 |
| Inside-Number | Number | 1 | 3 | 3 | 3559092 | 26476544 |
| Inside-Number | Inside | 2 | 1 | 4 | 5255315 | 19398656 |
| Inside-Number | Inside | 2 | 2 | 4 | 5253035 | 19398656 |
| Inside-Number | Inside | 2 | 3 | 4 | 5151024 | 19398656 |
| Inside-Number | Number | 2 | 1 | 4 | 3149872 | 19398656 |
| Inside-Number | Number | 2 | 2 | 4 | 2765010 | 19398656 |
| Inside-Number | Number | 2 | 3 | 4 | 2675994 | 19398656 |
| Inside-Number | Inside | 3 | 1 | 3 | 920241 | 26476544 |
| Inside-Number | Inside | 3 | 2 | 3 | 853147 | 26476544 |
| Inside-Number | Inside | 3 | 3 | 3 | 974109 | 26476544 |
| Inside-Number | Number | 3 | 1 | 4 | 5046083 | 19398656 |
| Inside-Number | Number | 3 | 2 | 4 | 5033831 | 19398656 |
| Inside-Number | Number | 3 | 3 | 4 | 5086233 | 19398656 |

Table 5: Resnet50 subnetwork sparsity statistics. Act. Param. is the number of active parameters in a subnetwork. Tot. Param. is the total number of parameters in the masked layers.

| Task | SR | Model # | Mask # | Stage | Act. Param. | Tot. Param. |
|---|---|---|---|---|---|---|
| Number-Contact | Contact | 1 | 1 | 3 | 1184224 | 26476544 |
| Number-Contact | Contact | 1 | 2 | 3 | 875632 | 26476544 |
| Number-Contact | Contact | 1 | 3 | 3 | 960475 | 26476544 |
| Number-Contact | Number | 1 | 1 | 4 | 703485 | 19398656 |
| Number-Contact | Number | 1 | 2 | 4 | 682724 | 19398656 |
| Number-Contact | Number | 1 | 3 | 4 | 684961 | 19398656 |
| Number-Contact | Contact | 2 | 1 | 3 | 833083 | 26476544 |
| Number-Contact | Contact | 2 | 2 | 3 | 935644 | 26476544 |
| Number-Contact | Contact | 2 | 3 | 3 | 903822 | 26476544 |
| Number-Contact | Number | 2 | 1 | 4 | 682249 | 19398656 |
| Number-Contact | Number | 2 | 2 | 4 | 660732 | 19398656 |
| Number-Contact | Number | 2 | 3 | 4 | 611524 | 19398656 |
| Number-Contact | Contact | 3 | 1 | 3 | 898919 | 26476544 |
| Number-Contact | Contact | 3 | 2 | 3 | 1129741 | 26476544 |
| Number-Contact | Contact | 3 | 3 | 3 | 765665 | 26476544 |
| Number-Contact | Number | 3 | 1 | 4 | 8734131 | 19398656 |
| Number-Contact | Number | 3 | 2 | 4 | 8880893 | 19398656 |
| Number-Contact | Number | 3 | 3 | 4 | 8900554 | 19398656 |
| Inside-Contact | Inside | 1 | 1 | 4 | 138289 | 19398656 |
| Inside-Contact | Inside | 1 | 2 | 4 | 102788 | 19398656 |
| Inside-Contact | Inside | 1 | 3 | 4 | 64056 | 19398656 |
| Inside-Contact | Contact | 1 | 1 | 4 | 687037 | 19398656 |
| Inside-Contact | Contact | 1 | 2 | 4 | 730767 | 19398656 |
| Inside-Contact | Contact | 1 | 3 | 4 | 594200 | 19398656 |
| Inside-Contact | Inside | 2 | 1 | 4 | 119585 | 19398656 |
| Inside-Contact | Inside | 2 | 2 | 4 | 153621 | 19398656 |
| Inside-Contact | Inside | 2 | 3 | 4 | 141847 | 19398656 |
| Inside-Contact | Contact | 2 | 1 | 4 | 880968 | 19398656 |
| Inside-Contact | Contact | 2 | 2 | 4 | 582672 | 19398656 |
| Inside-Contact | Contact | 2 | 3 | 4 | 549542 | 19398656 |
| Inside-Contact | Inside | 3 | 1 | 4 | 444801 | 19398656 |
| Inside-Contact | Inside | 3 | 2 | 4 | 404687 | 19398656 |
| Inside-Contact | Inside | 3 | 3 | 4 | 388647 | 19398656 |
| Inside-Contact | Contact | 3 | 1 | 4 | 2726236 | 19398656 |
| Inside-Contact | Contact | 3 | 2 | 4 | 2712782 | 19398656 |
| Inside-Contact | Contact | 3 | 3 | 4 | 2704681 | 19398656 |
| Inside-Number | Inside | 1 | 1 | 3 | 811096 | 26476544 |
| Inside-Number | Inside | 1 | 2 | 3 | 849964 | 26476544 |
| Inside-Number | Inside | 1 | 3 | 3 | 781551 | 26476544 |
| Inside-Number | Number | 1 | 1 | 0 | 14878394 | 27911360 |
| Inside-Number | Number | 1 | 2 | 0 | 14739139 | 27911360 |
| Inside-Number | Number | 1 | 3 | 0 | 14919954 | 27911360 |
| Inside-Number | Inside | 2 | 1 | 4 | 375073 | 19398656 |
| Inside-Number | Inside | 2 | 2 | 4 | 306550 | 19398656 |
| Inside-Number | Inside | 2 | 3 | 4 | 314610 | 19398656 |
| Inside-Number | Number | 2 | 1 | 3 | 2468331 | 26476544 |
| Inside-Number | Number | 2 | 2 | 3 | 3106382 | 26476544 |
| Inside-Number | Number | 2 | 3 | 3 | 3122791 | 26476544 |
| Inside-Number | Inside | 3 | 1 | 0 | 660344 | 27911360 |
| Inside-Number | Inside | 3 | 2 | 0 | 714676 | 27911360 |
| Inside-Number | Inside | 3 | 3 | 0 | 692889 | 27911360 |
| Inside-Number | Number | 3 | 1 | 3 | 3369673 | 26476544 |
| Inside-Number | Number | 3 | 2 | 3 | 3661479 | 26476544 |
| Inside-Number | Number | 3 | 3 | 3 | 3278225 | 26476544 |

Table 6: Resnet50 + SimCLR Subnetwork sparsity statistics. Act. Param. is the number of active parameters in a subnetwork. Tot. Param. is the total number of parameters in the masked layers.

| Task | SR | Model # | Mask # | Stage | Act. Param. | Tot. Param. |
|---|---|---|---|---|---|---|
| NUMBER-CONTACT | CONTACT | 1 | 1 | 4 | 12415313 | 46399488 |
| NUMBER-CONTACT | CONTACT | 1 | 2 | 4 | 11886093 | 46399488 |
| NUMBER-CONTACT | CONTACT | 1 | 3 | 4 | 11994752 | 46399488 |
| NUMBER-CONTACT | NUMBER | 1 | 1 | 0 | 364048 | 71222464 |
| NUMBER-CONTACT | NUMBER | 1 | 2 | 0 | 356776 | 71222464 |
| NUMBER-CONTACT | NUMBER | 1 | 3 | 0 | 430018 | 71222464 |
| NUMBER-CONTACT | CONTACT | 2 | 1 | 4 | 8156499 | 46399488 |
| NUMBER-CONTACT | CONTACT | 2 | 2 | 4 | 8188321 | 46399488 |
| NUMBER-CONTACT | CONTACT | 2 | 3 | 4 | 8730014 | 46399488 |
| NUMBER-CONTACT | NUMBER | 2 | 1 | 0 | 489546 | 71222464 |
| NUMBER-CONTACT | NUMBER | 2 | 2 | 0 | 479722 | 71222464 |
| NUMBER-CONTACT | NUMBER | 2 | 3 | 0 | 405563 | 71222464 |
| NUMBER-CONTACT | CONTACT | 3 | 1 | 4 | 11238193 | 46399488 |
| NUMBER-CONTACT | CONTACT | 3 | 2 | 4 | 11246123 | 46399488 |
| NUMBER-CONTACT | CONTACT | 3 | 3 | 4 | 11084672 | 46399488 |
| NUMBER-CONTACT | NUMBER | 3 | 1 | 4 | 792483 | 46399488 |
| NUMBER-CONTACT | NUMBER | 3 | 2 | 4 | 681226 | 46399488 |
| NUMBER-CONTACT | NUMBER | 3 | 3 | 4 | 834326 | 46399488 |
| INSIDE-CONTACT | INSIDE | 1 | 1 | 3 | 177339 | 67108864 |
| INSIDE-CONTACT | INSIDE | 1 | 2 | 3 | 147071 | 67108864 |
| INSIDE-CONTACT | INSIDE | 1 | 3 | 3 | 232306 | 67108864 |
| INSIDE-CONTACT | CONTACT | 1 | 1 | 3 | 1100205 | 67108864 |
| INSIDE-CONTACT | CONTACT | 1 | 2 | 3 | 966156 | 67108864 |
| INSIDE-CONTACT | CONTACT | 1 | 3 | 3 | 1043718 | 67108864 |
| INSIDE-CONTACT | INSIDE | 2 | 1 | 0 | 875532 | 71222464 |
| INSIDE-CONTACT | INSIDE | 2 | 2 | 0 | 493009 | 71222464 |
| INSIDE-CONTACT | INSIDE | 2 | 3 | 0 | 619362 | 71222464 |
| INSIDE-CONTACT | CONTACT | 2 | 1 | 3 | 5898635 | 67108864 |
| INSIDE-CONTACT | CONTACT | 2 | 2 | 3 | 6213208 | 67108864 |
| INSIDE-CONTACT | CONTACT | 2 | 3 | 3 | 5909038 | 67108864 |
| INSIDE-CONTACT | INSIDE | 3 | 1 | 3 | 557265 | 67108864 |
| INSIDE-CONTACT | INSIDE | 3 | 2 | 3 | 330289 | 67108864 |
| INSIDE-CONTACT | INSIDE | 3 | 3 | 3 | 710769 | 67108864 |
| INSIDE-CONTACT | CONTACT | 3 | 1 | 3 | 4632439 | 67108864 |
| INSIDE-CONTACT | CONTACT | 3 | 2 | 3 | 4376935 | 67108864 |
| INSIDE-CONTACT | CONTACT | 3 | 3 | 3 | 4964646 | 67108864 |
| INSIDE-NUMBER | INSIDE | 1 | 1 | 3 | 2081068 | 67108864 |
| INSIDE-NUMBER | INSIDE | 1 | 2 | 3 | 2106222 | 67108864 |
| INSIDE-NUMBER | INSIDE | 1 | 3 | 3 | 2007091 | 67108864 |
| INSIDE-NUMBER | NUMBER | 1 | 1 | 4 | 3541726 | 46399488 |
| INSIDE-NUMBER | NUMBER | 1 | 2 | 4 | 3560812 | 46399488 |
| INSIDE-NUMBER | NUMBER | 1 | 3 | 4 | 3583605 | 46399488 |
| INSIDE-NUMBER | INSIDE | 2 | 1 | 0 | 2242819 | 71222464 |
| INSIDE-NUMBER | INSIDE | 2 | 2 | 0 | 1963701 | 71222464 |
| INSIDE-NUMBER | INSIDE | 2 | 3 | 0 | 1330134 | 71222464 |
| INSIDE-NUMBER | NUMBER | 2 | 1 | 3 | 5749408 | 67108864 |
| INSIDE-NUMBER | NUMBER | 2 | 2 | 3 | 5511381 | 67108864 |
| INSIDE-NUMBER | NUMBER | 2 | 3 | 3 | 5472792 | 67108864 |
| INSIDE-NUMBER | INSIDE | 3 | 1 | 0 | 808884 | 71222464 |
| INSIDE-NUMBER | INSIDE | 3 | 2 | 0 | 851288 | 71222464 |
| INSIDE-NUMBER | INSIDE | 3 | 3 | 0 | 731729 | 71222464 |
| INSIDE-NUMBER | NUMBER | 3 | 1 | 0 | 3487441 | 71222464 |
| INSIDE-NUMBER | NUMBER | 3 | 2 | 0 | 3528010 | 71222464 |
| INSIDE-NUMBER | NUMBER | 3 | 3 | 0 | 4330284 | 71222464 |

Table 7: Wide Resnet50 Subnetwork sparsity statistics. Act. Param. is the number of active parameters in a subnetwork. Tot. Param. is the total number of parameters in the masked layers.

| Task | SR | Model # | Mask # | Stage | Act. Param. | Tot. Param. |
|------|-----|---------|--------|-------|-------------|-------------|
| (S) SV Agreement | Subject | 1 | 1 | 0 | 1246922 | 12845056 |
| (S) SV Agreement | Subject | 1 | 2 | 0 | 1242347 | 12845056 |
| (S) SV Agreement | Subject | 1 | 3 | 0 | 2128824 | 12845056 |
| (S) SV Agreement | Verb | 1 | 1 | 0 | 1668369 | 12845056 |
| (S) SV Agreement | Verb | 1 | 2 | 0 | 2600240 | 12845056 |
| (S) SV Agreement | Verb | 1 | 3 | 0 | 3353398 | 12845056 |
| (S) SV Agreement | Subject | 2 | 1 | 0 | 2728632 | 12845056 |
| (S) SV Agreement | Subject | 2 | 2 | 0 | 2663842 | 12845056 |
| (S) SV Agreement | Subject | 2 | 3 | 0 | 2681724 | 12845056 |
| (S) SV Agreement | Verb | 2 | 1 | 0 | 951132 | 12845056 |
| (S) SV Agreement | Verb | 2 | 2 | 0 | 1044003 | 12845056 |
| (S) SV Agreement | Verb | 2 | 3 | 0 | 1084848 | 12845056 |
| (S) SV Agreement | Subject | 3 | 1 | 0 | 328484 | 12845056 |
| (S) SV Agreement | Subject | 3 | 2 | 0 | 720899 | 12845056 |
| (S) SV Agreement | Subject | 3 | 3 | 0 | 323764 | 12845056 |
| (S) SV Agreement | Verb | 3 | 1 | 3 | 1794939 | 6553600 |
| (S) SV Agreement | Verb | 3 | 2 | 3 | 1702597 | 6553600 |
| (S) SV Agreement | Verb | 3 | 3 | 3 | 1567156 | 6553600 |
| (P) SV Agreement | Subject | 1 | 1 | 0 | 69748 | 12845056 |
| (P) SV Agreement | Subject | 1 | 2 | 0 | 68641 | 12845056 |
| (P) SV Agreement | Subject | 1 | 3 | 0 | 52336 | 12845056 |
| (P) SV Agreement | Verb | 1 | 1 | 3 | 640889 | 6553600 |
| (P) SV Agreement | Verb | 1 | 2 | 3 | 477101 | 6553600 |
| (P) SV Agreement | Verb | 1 | 3 | 3 | 656202 | 6553600 |
| (P) SV Agreement | Subject | 2 | 1 | 0 | 119037 | 12845056 |
| (P) SV Agreement | Subject | 2 | 2 | 0 | 125594 | 12845056 |
| (P) SV Agreement | Subject | 2 | 3 | 0 | 122828 | 12845056 |
| (P) SV Agreement | Verb | 2 | 1 | 0 | 215555 | 12845056 |
| (P) SV Agreement | Verb | 2 | 2 | 0 | 202185 | 12845056 |
| (P) SV Agreement | Verb | 2 | 3 | 0 | 56134 | 12845056 |
| (P) SV Agreement | Subject | 3 | 1 | 0 | 86084 | 12845056 |
| (P) SV Agreement | Subject | 3 | 2 | 0 | 49694 | 12845056 |
| (P) SV Agreement | Subject | 3 | 3 | 0 | 166614 | 12845056 |
| (P) SV Agreement | Verb | 3 | 1 | 0 | 149398 | 12845056 |
| (P) SV Agreement | Verb | 3 | 2 | 0 | 221074 | 12845056 |
| (P) SV Agreement | Verb | 3 | 3 | 0 | 133149 | 12845056 |

Table 8: BERT Subject-Verb Agreement Subnetwork sparsity statistics. Act. Param. is the number of active parameters in a subnetwork. Tot. Param. is the total number of parameters in the masked layers.

models trained on compositional tasks. In Section 7, each (model, subroutine) pair had its own set of masking hyperparameters. We use these same hyperparameters for each (randomly-intialized model, subroutine) pair in order to make the results as comparable as possible.

In Figure 16 and 17, we observe that masking random networks produces distinctly different patterns of results than we presented in Section 7. Though it is possible find a subnetwork that computes a target subroutine and not the other subroutine, the ablation results do not follow the pattern that one would expect of a compositional model. This accords with Ramanujan et al. (2020), which demonstrates that training a binary mask over a randomly-weighted network can still produce performant subnetworks. The ablation results indicate that these subnetworks are not causally implicated in model behavior. In the case of Resnet50 (Figure 16, Bottom) we observe that ablating the learned subnetworks collapses performance to chance for all tasks. In the case of BERT-Small (Figure 17, Bottom), we observe that ablating the learned subnetworks oftentimes yields high performance on **Test Target Subroutine** and low performance on **Test Other Subroutine**, which is the *opposite* of the expected trend for a compositional model. Thus, we can be more confident that

| TASK | SR | MODEL # | MASK # | STAGE | ACT. PARAM. | TOT. PARAM. |
|------|----|---------|--------|-------|-------------|-------------|
| (S) ANAPHORA | PRONOUN | 1 | 1 | 0 | 370568 | 12845056 |
| (S) ANAPHORA | PRONOUN | 1 | 2 | 0 | 372203 | 12845056 |
| (S) ANAPHORA | PRONOUN | 1 | 3 | 0 | 369123 | 12845056 |
| (S) ANAPHORA | ANTECEDENT | 1 | 1 | 0 | 169120 | 12845056 |
| (S) ANAPHORA | ANTECEDENT | 1 | 2 | 0 | 150554 | 12845056 |
| (S) ANAPHORA | ANTECEDENT | 1 | 3 | 0 | 231819 | 12845056 |
| (S) ANAPHORA | PRONOUN | 2 | 1 | 0 | 968021 | 12845056 |
| (S) ANAPHORA | PRONOUN | 2 | 2 | 0 | 971063 | 12845056 |
| (S) ANAPHORA | PRONOUN | 2 | 3 | 0 | 970910 | 12845056 |
| (S) ANAPHORA | ANTECEDENT | 2 | 1 | 0 | 108544 | 12845056 |
| (S) ANAPHORA | ANTECEDENT | 2 | 2 | 0 | 110871 | 12845056 |
| (S) ANAPHORA | ANTECEDENT | 2 | 3 | 0 | 108347 | 12845056 |
| (S) ANAPHORA | PRONOUN | 3 | 1 | 0 | 79069 | 12845056 |
| (S) ANAPHORA | PRONOUN | 3 | 2 | 0 | 79031 | 12845056 |
| (S) ANAPHORA | PRONOUN | 3 | 3 | 0 | 82788 | 12845056 |
| (S) ANAPHORA | ANTECEDENT | 3 | 1 | 0 | 1854552 | 12845056 |
| (S) ANAPHORA | ANTECEDENT | 3 | 2 | 0 | 1848327 | 12845056 |
| (S) ANAPHORA | ANTECEDENT | 3 | 3 | 0 | 1874968 | 12845056 |
| (P) ANAPHORA | PRONOUN | 1 | 1 | 0 | 325597 | 12845056 |
| (P) ANAPHORA | PRONOUN | 1 | 2 | 0 | 417498 | 12845056 |
| (P) ANAPHORA | PRONOUN | 1 | 3 | 0 | 642805 | 12845056 |
| (P) ANAPHORA | ANTECEDENT | 1 | 1 | 0 | 286739 | 12845056 |
| (P) ANAPHORA | ANTECEDENT | 1 | 2 | 0 | 336572 | 12845056 |
| (P) ANAPHORA | ANTECEDENT | 1 | 3 | 0 | 405887 | 12845056 |
| (P) ANAPHORA | PRONOUN | 2 | 1 | 0 | 24818 | 12845056 |
| (P) ANAPHORA | PRONOUN | 2 | 2 | 0 | 24805 | 12845056 |
| (P) ANAPHORA | PRONOUN | 2 | 3 | 0 | 24855 | 12845056 |
| (P) ANAPHORA | ANTECEDENT | 2 | 1 | 3 | 1154161 | 6553600 |
| (P) ANAPHORA | ANTECEDENT | 2 | 2 | 3 | 1183436 | 6553600 |
| (P) ANAPHORA | ANTECEDENT | 2 | 3 | 3 | 1159462 | 6553600 |
| (P) ANAPHORA | PRONOUN | 3 | 1 | 0 | 144186 | 12845056 |
| (P) ANAPHORA | PRONOUN | 3 | 2 | 0 | 151531 | 12845056 |
| (P) ANAPHORA | PRONOUN | 3 | 3 | 0 | 153897 | 12845056 |
| (P) ANAPHORA | ANTECEDENT | 3 | 1 | 0 | 4606842 | 12845056 |
| (P) ANAPHORA | ANTECEDENT | 3 | 2 | 0 | 5134758 | 12845056 |
| (P) ANAPHORA | ANTECEDENT | 3 | 3 | 0 | 5041888 | 12845056 |

Table 9: BERT Anaphora Subnetwork sparsity statistics. Act. Param. is the number of active parameters in a subnetwork. Tot. Param. is the total number of parameters in the masked layers.

the main results presented in Section 7 reflect the internal mechanisms of trained models, and are not epiphenomenal artifacts of training binary masks over networks.

## I.1 Statistical Analysis of Main Results vs. Random Results

In order to assess whether the results given by random models are significantly different from our main results, we fit a generalized linear model (GLM) with robust clustered standard errors for each combination of model architecture, compositional task, and subroutine. This GLM includes a dummy variable indicating whether the results are from a subnetwork or an ablated model, a dummy variable indicating whether the base model is trained or random, and it clusters observations by base model. For language experiments, we collapse across singular and plural instances of the same subroutine. The coefficient of the Trained vs. Random dummy variable in this model assesses whether the performance of the discovered subnetworks are significantly different in the trained and random conditions. From this model, we can also perform a linear hypothesis test to assess whether ablated model performance is significantly different between the trained and random conditions. Table 12 provides the relevant statistics. Across the board, we see that there is often a significant difference

| Task | SR | Model # | Mask # | Stage | Act. Param. | Tot. Param. |
|---|---|---|---|---|---|---|
| (S) SV Agreement | Subject | 1 | 1 | 0 | 59736 | 12845056 |
| (S) SV Agreement | Subject | 1 | 2 | 0 | 839770 | 12845056 |
| (S) SV Agreement | Subject | 1 | 3 | 0 | 620869 | 12845056 |
| (S) SV Agreement | Verb | 1 | 1 | 3 | 65229 | 6553600 |
| (S) SV Agreement | Verb | 1 | 2 | 3 | 65356 | 6553600 |
| (S) SV Agreement | Verb | 1 | 3 | 3 | 65220 | 6553600 |
| (S) SV Agreement | Subject | 2 | 1 | 0 | 511359 | 12845056 |
| (S) SV Agreement | Subject | 2 | 2 | 0 | 70725 | 12845056 |
| (S) SV Agreement | Subject | 2 | 3 | 0 | 555174 | 12845056 |
| (S) SV Agreement | Verb | 2 | 1 | 0 | 16940 | 12845056 |
| (S) SV Agreement | Verb | 2 | 2 | 0 | 39218 | 12845056 |
| (S) SV Agreement | Verb | 2 | 3 | 0 | 19321 | 12845056 |
| (S) SV Agreement | Subject | 3 | 1 | 0 | 8514 | 12845056 |
| (S) SV Agreement | Subject | 3 | 2 | 0 | 8613 | 12845056 |
| (S) SV Agreement | Subject | 3 | 3 | 0 | 8593 | 12845056 |
| (S) SV Agreement | Verb | 3 | 1 | 3 | 64969 | 6553600 |
| (S) SV Agreement | Verb | 3 | 2 | 3 | 65098 | 6553600 |
| (S) SV Agreement | Verb | 3 | 3 | 3 | 64955 | 6553600 |
| (P) SV Agreement | Subject | 1 | 1 | 0 | 45792 | 12845056 |
| (P) SV Agreement | Subject | 1 | 2 | 0 | 45527 | 12845056 |
| (P) SV Agreement | Subject | 1 | 3 | 0 | 45616 | 12845056 |
| (P) SV Agreement | Verb | 1 | 1 | 0 | 16800 | 12845056 |
| (P) SV Agreement | Verb | 1 | 2 | 0 | 24013 | 12845056 |
| (P) SV Agreement | Verb | 1 | 3 | 0 | 23988 | 12845056 |
| (P) SV Agreement | Subject | 2 | 1 | 0 | 47651 | 12845056 |
| (P) SV Agreement | Subject | 2 | 2 | 0 | 47502 | 12845056 |
| (P) SV Agreement | Subject | 2 | 3 | 0 | 47811 | 12845056 |
| (P) SV Agreement | Verb | 2 | 1 | 3 | 100005 | 6553600 |
| (P) SV Agreement | Verb | 2 | 2 | 3 | 100100 | 6553600 |
| (P) SV Agreement | Verb | 2 | 3 | 3 | 100029 | 6553600 |
| (P) SV Agreement | Subject | 3 | 1 | 3 | 81133 | 6553600 |
| (P) SV Agreement | Subject | 3 | 2 | 3 | 81203 | 6553600 |
| (P) SV Agreement | Subject | 3 | 3 | 3 | 81218 | 6553600 |
| (P) SV Agreement | Verb | 3 | 1 | 0 | 15302 | 12845056 |
| (P) SV Agreement | Verb | 3 | 2 | 0 | 9423 | 12845056 |
| (P) SV Agreement | Verb | 3 | 3 | 0 | 15900 | 12845056 |

Table 10: BERT + LM Subject-Verb Agreement Subnetwork sparsity statistics. Act. Param. is the number of active parameters in a subnetwork. Tot. Param. is the total number of parameters in the masked layers.

between ablating the discovered subnetworks in random vs. trained models, even with a small sample size.

## J   Pruned Model Analysis

In this section, we analyze the structural compositionality of models after they have been pruned. We analyze the impact of pruning on structural compositionality on Resnet50 models trained on Number-Contact. We use continuous sparsification to train binary masks over the three Number-Contact Resnet50 models analyzed in the main paper, resulting in three sparse models that perform well on the Number-Contact task. We search over mask initialization and learning rate hyperparameters to find the best pruning configuration for each model. Then, we run the same structural compositionality analysis described in the main paper. We see from Figure 18 that the results on the pruned Resnet50 models closely resemble those from the full Resnet50 models.

| Task | SR | Model # | Mask # | Stage | Act. Param. | Tot. Param. |
|---|---|---|---|---|---|---|
| (S) Anaphora | Pronoun | 1 | 1 | 0 | 6590452 | 12845056 |
| (S) Anaphora | Pronoun | 1 | 2 | 0 | 6702195 | 12845056 |
| (S) Anaphora | Pronoun | 1 | 3 | 0 | 6519670 | 12845056 |
| (S) Anaphora | Antecedent | 1 | 1 | 0 | 832044 | 12845056 |
| (S) Anaphora | Antecedent | 1 | 2 | 0 | 833149 | 12845056 |
| (S) Anaphora | Antecedent | 1 | 3 | 0 | 835278 | 12845056 |
| (S) Anaphora | Pronoun | 2 | 1 | 3 | 193677 | 6553600 |
| (S) Anaphora | Pronoun | 2 | 2 | 3 | 198612 | 6553600 |
| (S) Anaphora | Pronoun | 2 | 3 | 3 | 198956 | 6553600 |
| (S) Anaphora | Antecedent | 2 | 1 | 0 | 39355 | 12845056 |
| (S) Anaphora | Antecedent | 2 | 2 | 0 | 27784 | 12845056 |
| (S) Anaphora | Antecedent | 2 | 3 | 0 | 34674 | 12845056 |
| (S) Anaphora | Pronoun | 3 | 1 | 0 | 36495 | 12845056 |
| (S) Anaphora | Pronoun | 3 | 2 | 0 | 1458843 | 12845056 |
| (S) Anaphora | Pronoun | 3 | 3 | 0 | 75100 | 12845056 |
| (S) Anaphora | Antecedent | 3 | 1 | 0 | 648936 | 12845056 |
| (S) Anaphora | Antecedent | 3 | 2 | 0 | 680218 | 12845056 |
| (S) Anaphora | Antecedent | 3 | 3 | 0 | 760770 | 12845056 |
| (P) Anaphora | Pronoun | 1 | 1 | 0 | 11374 | 12845056 |
| (P) Anaphora | Pronoun | 1 | 2 | 0 | 11251 | 12845056 |
| (P) Anaphora | Pronoun | 1 | 3 | 0 | 11369 | 12845056 |
| (P) Anaphora | Antecedent | 1 | 1 | 0 | 1152444 | 12845056 |
| (P) Anaphora | Antecedent | 1 | 2 | 0 | 1152518 | 12845056 |
| (P) Anaphora | Antecedent | 1 | 3 | 0 | 1149693 | 12845056 |
| (P) Anaphora | Pronoun | 2 | 1 | 0 | 11327 | 12845056 |
| (P) Anaphora | Pronoun | 2 | 2 | 0 | 11321 | 12845056 |
| (P) Anaphora | Pronoun | 2 | 3 | 0 | 11275 | 12845056 |
| (P) Anaphora | Antecedent | 2 | 1 | 0 | 43274 | 12845056 |
| (P) Anaphora | Antecedent | 2 | 2 | 0 | 44220 | 12845056 |
| (P) Anaphora | Antecedent | 2 | 3 | 0 | 45632 | 12845056 |
| (P) Anaphora | Pronoun | 3 | 1 | 0 | 28866 | 12845056 |
| (P) Anaphora | Pronoun | 3 | 2 | 0 | 28887 | 12845056 |
| (P) Anaphora | Pronoun | 3 | 3 | 0 | 29101 | 12845056 |
| (P) Anaphora | Antecedent | 3 | 1 | 0 | 4292104 | 12845056 |
| (P) Anaphora | Antecedent | 3 | 2 | 0 | 4350952 | 12845056 |
| (P) Anaphora | Antecedent | 3 | 3 | 0 | 4048117 | 12845056 |

Table 11: BERT + LM Anaphora Subnetwork sparsity statistics. Act. Param. is the number of active parameters in a subnetwork. Tot. Param. is the total number of parameters in the masked layers.

| Comp. Task | Model | SR | Random Coef. Z | Linear Hypothesis $\chi^2$ |
|---|---|---|---|---|
| In. Cont. | RN50 | Inside | -1.36 | 2.44 |
| In. Cont. | RN50 | Contact | -1.73. | 1.17 |
| In. Num. | RN50 | Inside | -1.92. | 29.20*** |
| In. Num. | RN50 | Number | -1.41 | 48.95*** |
| Cont. Num. | RN50 | Contact | -0.51 | 57.47*** |
| Cont. Num. | RN50 | Number | -0.26 | 0.54 |
| SV Agr | BERT | Subj. | 2.13* | 19.17*** |
| SV Agr | BERT | Verb | 2.61** | 70.08*** |
| Anaphora | BERT | Pronoun | 2.46* | 122.64*** |
| Anaphora | BERT | Antecedent | 0.24 | 17.87*** |

Table 12: Statistics from two factor GLM with robust clustered standard errors. . indicates significance at p = .1, * indicates significance at p = .05, ** indicates significance at p = .01, *** indicates significance at p = .001

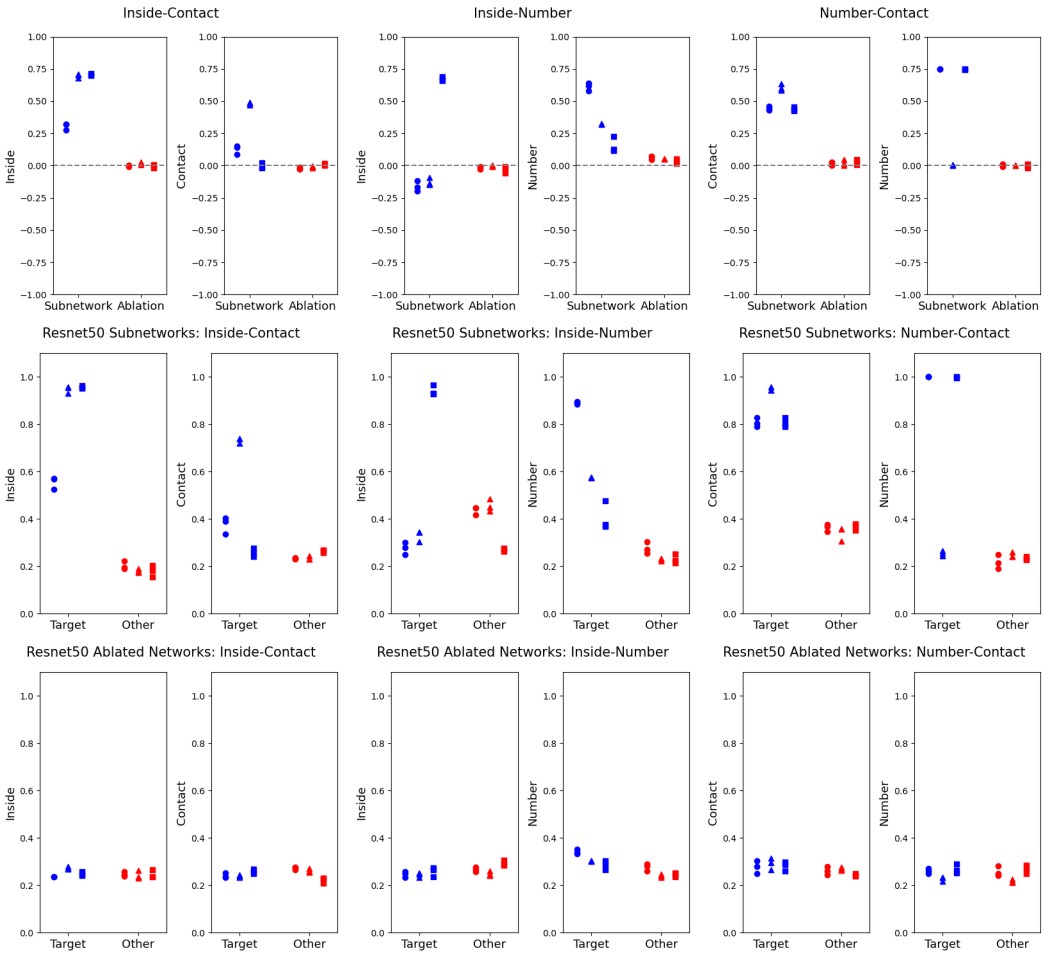

Figure 16: Results from training masks over a randomly-initialized Resnet50. (Top) Plots displaying differences in performance. (Middle) Plots displaying subnetwork performance on each task. (Bottom) Plots displaying ablated model performance on each task. Across the board, we see that masks over random networks can produce subnetworks that achieve better accuracy on on **Test Target Subroutine** than on **Test Other Subroutine**, but that ablating these subnetworks results in (equally) poor performance on both of these datasets.

## K Subnetwork Overlap Analysis

In this section, we analyze the overlap in the subnetworks that were discovered within the same base model. We perform this analysis on one model, a Resnet50 trained on the Inside-Number task. From Figure 4, we see that this model appears to exhibit structural compositionality. We analyze this model because subnetwork masking started at the same layer for both subroutines, which allows for a straightforward comparison of the overlap between subnetworks. All results are shown in Figure 19.

Following previous work (Csordás et al., 2021), we compute the per-layer intersection over union (IoU) to quanitify subnetwork overlap. First, we do this for each of the three subnetworks discovered for the Inside subroutine. See these results in Table 13. Next, we compute the same for the three subnetworks discovered for the Number subroutine. See these results in Table 14. The Inside subroutine gives near ceiling agreement, while the Number task exhibits much lower agreement. This indicates that the subnetworks we uncover are somewhat noisy for the Number subroutine, but not for the Inside subroutine. Finally, we compute the intersection of each set of subnetworks, and compute the per-layer intersection over union *between* tasks. See Table 15. We see very low agreement between tasks, especially before the final MLP. Notably, this between-task agreement is consistently

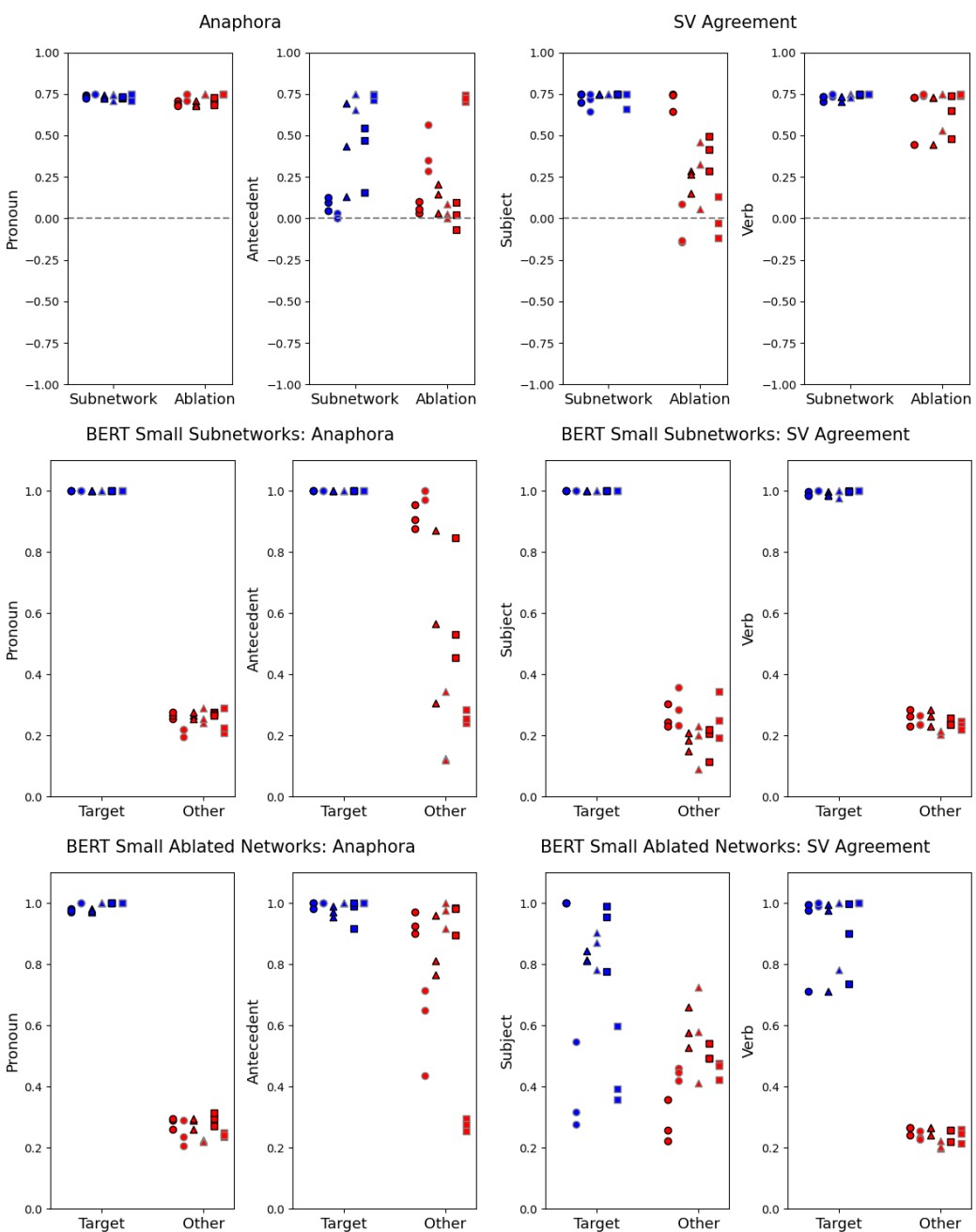

Figure 17: Results from training masks over a randomly-initialized BERT-Small. (Top) Plots displaying differences in performance. (Middle) Plots displaying subnetwork performance on each task. (Bottom) Plots displaying ablated model performance on each task. Across the board, we see that masks over random networks can produce subnetworks that achieve better accuracy on on **Test Target Subroutine** than on **Test Other Subroutine**. Surprisingly, ablating these subnetworks still results in better accuracy on on **Test Target Subroutine** than on **Test Other Subroutine**.

lower than both within-task agreement values for each layer. This reinforces our interpretation that these subnetworks are organized in a modular fashion within the base models.

| LAYER | IoU |
|---|---|
| BACKBONE.LAYER4.0.CONV1 | 0.974 |
| BACKBONE.LAYER4.0.CONV2 | 0.963 |
| BACKBONE.LAYER4.0.CONV3 | 0.971 |
| BACKBONE.LAYER4.0.DOWNSAMPLE.0 | 0.971 |
| BACKBONE.LAYER4.1.CONV1 | 0.970 |
| BACKBONE.LAYER4.1.CONV2 | 0.960 |
| BACKBONE.LAYER4.1.CONV3 | 0.969 |
| BACKBONE.LAYER4.2.CONV1 | 0.975 |
| BACKBONE.LAYER4.2.CONV2 | 0.971 |
| BACKBONE.LAYER4.2.CONV3 | 0.970 |
| MLP.MODEL.0 | 0.994 |
| MLP.MODEL.2 | 0.989 |

Table 13: IoU computed over the three discovered subnetworks in for the Inside subroutine

| LAYER | IoU |
|---|---|
| BACKBONE.LAYER4.0.CONV1 | 0.370 |
| BACKBONE.LAYER4.0.CONV2 | 0.236 |
| BACKBONE.LAYER4.0.CONV3 | 0.264 |
| BACKBONE.LAYER4.0.DOWNSAMPLE.0 | 0.257 |
| BACKBONE.LAYER4.1.CONV1 | 0.287 |
| BACKBONE.LAYER4.1.CONV2 | 0.208 |
| BACKBONE.LAYER4.1.CONV3 | 0.218 |
| BACKBONE.LAYER4.2.CONV1 | 0.197 |
| BACKBONE.LAYER4.2.CONV2 | 0.129 |
| BACKBONE.LAYER4.2.CONV3 | 0.162 |
| MLP.MODEL.0 | 0.516 |
| MLP.MODEL.2 | 0.411 |

Table 14: IoU computed over the three discovered subnetworks in for the Number subroutine

| LAYER | IoU |
|---|---|
| BACKBONE.LAYER4.0.CONV1 | 0.122 |
| BACKBONE.LAYER4.0.CONV2 | 0.062 |
| BACKBONE.LAYER4.0.CONV3 | 0.087 |
| BACKBONE.LAYER4.0.DOWNSAMPLE.0 | 0.057 |
| BACKBONE.LAYER4.1.CONV1 | 0.070 |
| BACKBONE.LAYER4.1.CONV2 | 0.054 |
| BACKBONE.LAYER4.1.CONV3 | 0.076 |
| BACKBONE.LAYER4.2.CONV1 | 0.034 |
| BACKBONE.LAYER4.2.CONV2 | 0.027 |
| BACKBONE.LAYER4.2.CONV3 | 0.055 |
| MLP.MODEL.0 | 0.237 |
| MLP.MODEL.2 | 0.321 |

Table 15: IoU computed over the *intersections* of the subnetworks discovered for each task.

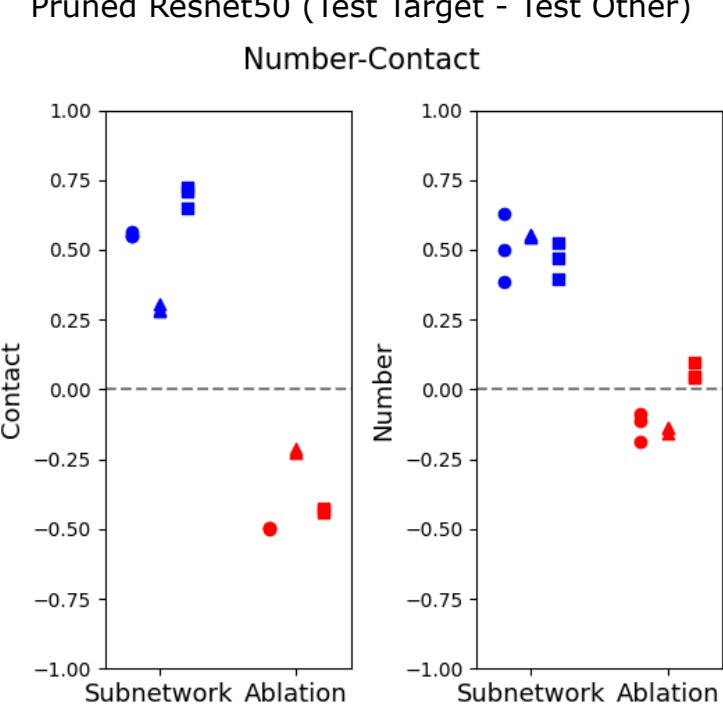

Figure 18: Structural compositionality analysis using pruned Resnet50 models on the Number-Contact task. We see that these results closely match those found in the main paper.

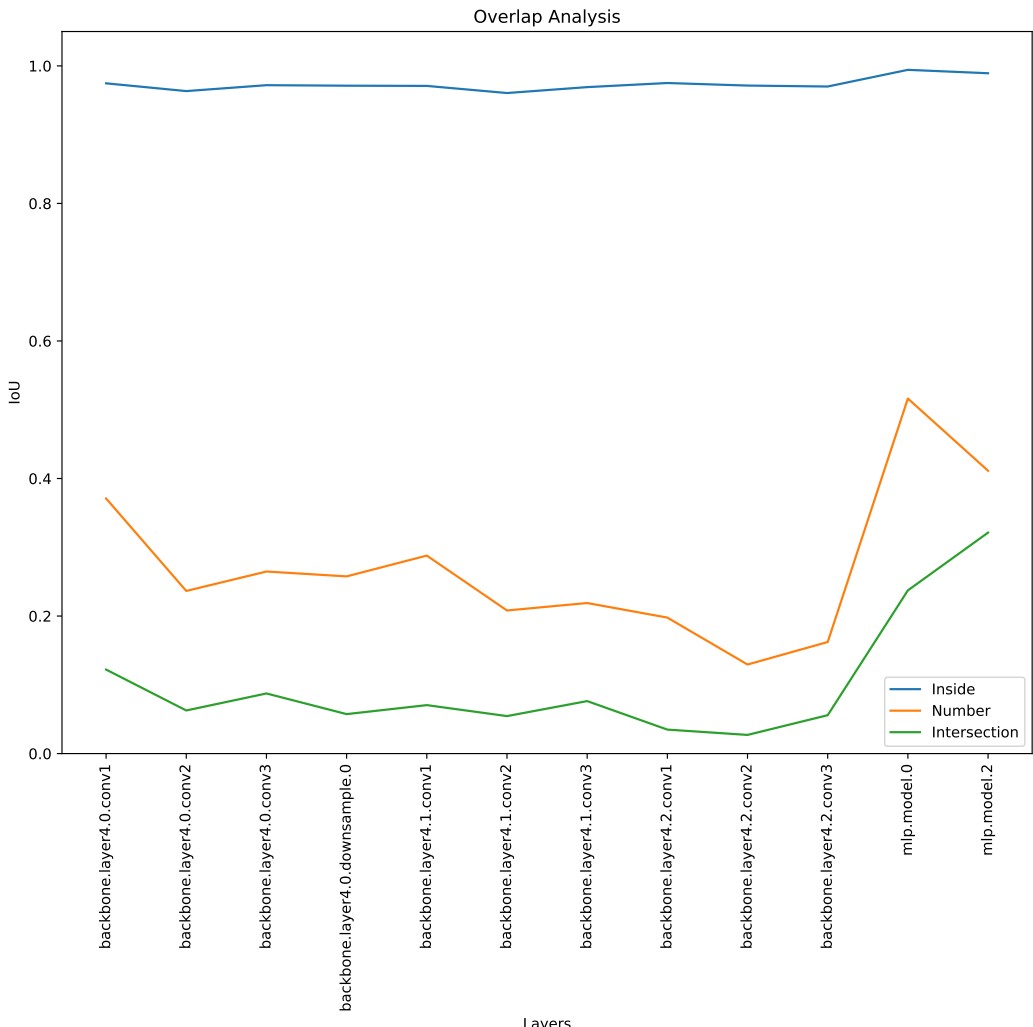

Figure 19: The results of subnetwork overlap analysis for one base model trained on the Inside-Number task. These results show that within-task subnetwork overlap is substantially higher than between-task subnetwork overlap, measured by intersection over union (IoU). Within a task, we are computing the IoU between 3 subnetworks trained for that task. Between tasks, we first take the intersection of all 3 subnetworks trained for each task, and then compute the IoU of the intersections.

