# OpenReview forum: "Break It Down:  Evidence for Structural Compositionality in Neural Networks"
_NeurIPS.cc/2023/Conference — NeurIPS 2023 spotlight_

### Official Review · Reviewer_DSTn · 2023-06-24

**Soundness:** 3 good
**Presentation:** 4 excellent
**Contribution:** 4 excellent
**Rating:** 7
**Confidence:** 4

**Summary:**

This paper studies the structural compositional problem is a novel perspective. It first defines structural compositionality as the extent to which neural networks decompose a complex task into a series of subroutines and implement them modularly. Then, the paper designs several clever experiments to show that many models indeed implement subroutines in their subnetworks, which demystifies how the compositional generalization ability might occur in deep neural networks. Although the paper could be further improved in the following directions (see the limitation part), I think the current version already meets the criteria of NeurIPS and can inspire the community a lot. So I would give an accepting score to the current version. In summary, the paper is easy to follow and quite novel to me. I enjoy reading it a lot.

**Strengths:**

1. Rather than the downstream generalization ability or the data attributes, the paper focuses on the composition of different rules (functions, or subroutines), which is quite novel and persuasive.
2. The experimental designs are quite ingenious and persuasive to me, it relates the subroutines to subnetworks using the mask function on each parameter.
3. The paper investigates different model architectures (ResNet, ViT, BERT) and different input signals (image and language).

**Weaknesses:**

See the limitation part.

**Questions:**

1. For the non-compositional solution panel in Figure 1, it might be better to draw class A and class B in the upper right and lower left corners respectively. Then, adding a decision boundary (i.e., the diagonal) would be helpful.
2. It is a little hard for me to understand what the bottom-right panel of Figure 1 is talking about before reading the experimental part.
3. Seems that the filling color of the four panels in the bottom-right part of Figure 2 is wrong. IIUC, the color for the ablation model’s performance would be the inverse of those for the subnetwork.
4. In section 7, the authors mentioned that ViT fails on the proposed problem, is there any possible explanation?
5. For Figure 4, adding the legends for *** and gray dots would be helpful.

**Limitations:**

I believe solving the following two concerns could make the paper stronger and bring more insights to the community (it is also fine to consider them in future works).

1. The paper selects the subnetwork by training a mask function $m$ and generates two subnetworks, i.e., $Sub_i$ and $M_{ablate}$, based on it. Then it might be interesting to draw the mask for different subnets (keeping the structure of the model) to see if there exist any interesting patterns. For example, it is possible that these two complementary networks are using different halves of the neurons across all layers; or they might share some lower layers but split apart in the higher layers. I guess these results might be quite helpful for us to understand HOW different subroutines are stored in different model architectures. (Maybe whether the model is pretrained also influences this split.)
2. Besides the related works mentioned in the paper, there is also another line of works discussing how compositional mapping emerges in human language and neural network representations. It is named iterated learning, which is first proposed in the field of language evolution to explain why modern human language is so structural [1] and then extended to the deep learning conditions, e.g., emergent communication in [2] and visual question-answering in [3]. It might be interesting to consider the relationships between these works.

[1] Kirby, Simon, et al. "Cumulative cultural evolution in the laboratory: An experimental approach to the origins of structure in human language." PNAS 2008

[2] Ren, Yi, et al. "Compositional languages emerge in a neural iterated learning model." ICLR 2020

[3] Vani, Ankit, et al. "Iterated learning for emergent systematicity in VQA." ICLR 2021

---

> ### Author Rebuttal · Authors · 2023-08-09
>
> Thank you for your excellent feedback and questions about our work! We will edit the figures for clarity in the final version of the paper. We agree that a figure describing how subnetworks are distributed throughout a model would be a valuable addition to the paper, and are working on creating such a diagram for the final version. Figure 18 in the supplementary material contains this information for one model, but we agree that a better version of this figure should appear in the main text. Also, thank you for the references to the Iterated Learning literature - they are very cool and relevant! We will add a section to our Related Work describing how iterated learning can be used to encourage compositional behavior in neural networks.

---

> > ### Comment · Reviewer_DSTn · 2023-08-13
> > **Thank you for the rebuttal**
> >
> > Thank you for your response and for accepting my suggestion. I am looking forward to seeing the final version. This is really a cool paper.

---

### Official Review · Reviewer_oWEk · 2023-07-04

**Soundness:** 4 excellent
**Presentation:** 3 good
**Contribution:** 4 excellent
**Rating:** 8
**Confidence:** 4

**Summary:**

The paper investigates the concept of structural compositionality in neural networks - subnetworks that implement specific subroutines, such as computing the syntactic number of specific words in a sentence.

**Strengths:**

1. Conceptual modularity in neural networks has been an idea that has been studied and structurally implemented across various domains: explainability, reasoning, efficiency. The authors advance the field of study of modularity in neural networks by proposing the idea of structural compositionality, which is both is very well thought out and explained. The experimental plan is also carefully controlled to ensure causal observations regarding the existence of structural compositionality in networks across both vision and language tasks. Overall this is a very well done study.

**Weaknesses:**

No major waknesses in my opinion

**Questions:**

**Important Related Work:** While the authors mention the field of mechanistic interpretability, they fail to mention some key papers in this field that explore similar ideas as the authors Chugtai et al [1] reverse engineer small neural networks to show that they learn to implement group compositions for any finite group using partical novel algorithm predicted from theory. Nouha et al [2] show that transformers solve compositional tasks by approximating part of the full computational graph as linear sub-graph matching, and provide theoretical proofs on how increasing compositional task complexity would lead to degradation in performance. I think these papers should be referred so that readers will have a more complete picture of compositionality in neural networks.

### References

1. Chughtai, Bilal, Lawrence Chan, and Neel Nanda. "A Toy Model of Universality: Reverse Engineering how Networks Learn Group Operations." (2023).
2. Dziri, Nouha, et al. "Faith and Fate: Limits of Transformers on Compositionality." arXiv preprint arXiv:2305.18654 (2023).

**Limitations:**

The authors do a good job at highlighting the limitations (e.g. specifying subroutines in advance) of their work. I do not particularly see potential negative social impacts of this work as it mostly pertains to explainability of models.

---

> ### Author Rebuttal · Authors · 2023-08-09
>
> Thank you for your thoughtful comments on our work! The related works that you’ve referenced are extremely interesting, and will strengthen our discussion - thanks for pointing them out! We will add a new section to our discussion that describes the relationship between structural compositionality and the forms of compositionality that are discussed in these papers.

---

> > ### Comment · Reviewer_oWEk · 2023-08-10
> > **Response to rebuttal**
> >
> > Thank you for taking the time to write a rebuttal, and accepting my suggestion to include the additional references and discuss them wrt your contributions. I stand by my original rating, and look forward to (hopefully) seeing more work on mechanistic interpretability and structural compositionality based on this paper.

---

### Official Review · Reviewer_W9E9 · 2023-07-06

**Soundness:** 3 good
**Presentation:** 4 excellent
**Contribution:** 4 excellent
**Rating:** 7
**Confidence:** 4

**Summary:**

This research paper explores the concept of compositionality in neural networks, a contentious topic in the field of AI. Compositionality, which is a defining feature of human cognition, allows for abstract and flexible processing of language and visuals. The debate lies in whether neural networks need explicit symbolic systems to implement compositional solutions, or if these solutions can be implicitly learned during training. This research introduces "structural compositionality", which gauges the degree to which neural networks can break down compositional tasks into subroutines and implement them in a modular fashion. Surprisingly, the study found evidence that many models do implement subroutines in modular subnetworks. Additionally, it was found that unsupervised pretraining leads to a more consistently compositional structure in language models. This research contributes to mechanistic interpretability, helping explain the algorithms that neural networks implement in their weights, using techniques from model pruning.

**Strengths:**

The paper is quite interesting, timely and I think that most people would find the results surprising. The area of mechanistic interpretability is very important these days. One of the strongest points of this paper is the elegant approach it is taking, in terms of experimental design, to construct compositional tasks and to discover whether the presence of such "functional compositionality" also results in specific subnetworks for each "concept". The paper presents results both for images and for language experiments. The results are clear and the presentation of the paper is very good.

**Weaknesses:**

Of course, one general weakness is that the main result of the paper is through empirical results. The reader may not be convinced whether the claims of the paper are actually true in general -- or whether they are true only in some architectures and tasks.

Another major weakness, which the authors also identify as a limitation of their study, is that the one must know which subroutines to look for.

**Questions:**

A question for the authors to think about: what is the role of pruning (or network density, more generally) in the emergence of structural compositionality? Would you expect that emergence to be equally likely in a dense network and in a sparse network that has resulted through  pruning?

Another question to think about: what is the role of the network's depth in the emergence of structural compositionality? Functional compositionality can have its own depth. How is that functional depth relate to the depth of the network itself?

**Limitations:**

The paper is honest about its limitations. The important of them is that one has to know in advance which "subroutines" (I do not like this term actually -- maybe "concepts" or "subfunctions" would be better terns) to look for. This limitation however may be addressed by other work.

---

> ### Author Rebuttal · Authors · 2023-08-09
>
> Thank you for your excellent feedback and questions about our work! We agree that the current work is a purely empirical study, and so we attempted to demonstrate the effect on various architectures and domains to help convince the reader that structural compositionality is a property of a wide class of models. That said, since submitting (and even more so after your comments), we have been investigating possible theoretical work that might relate to our findings. Particularly interesting is this recent paper on compositionality and sparse representations in neural networks (Poggio 2023, https://cbmm.mit.edu/sites/default/files/publications/Theoretical_Framework__How_Deep_Nets_May_Work_24.pdf). We will add a new section to our discussion to outline the ways in which our results are consistent with such theoretical work, and point towards possible future experiments that could better connect the two.
>
> Your question regarding how sparsity interacts with structural compositionality is very interesting! We will run an experiment during the discussion period to investigate this, and update leave a comment on OpenReview with the results next week. We will also add these results to the Appendix.

---

> > ### Comment · Reviewer_W9E9 · 2023-08-10
> >
> > thanks for the thoughtful rebuttal. After considering the other reviews too, I have decided to stay with my original score.

---

> > > ### Author Response · Authors · 2023-08-20
> > >
> > > Thanks! Just to follow up regarding the sparsity experiment: we pruned each Resnet50 used in our Number-Contact vision experiments, and then reran the experiment in an identical manner on the pruned models. We do not observe any salient differences between these results and our original results. In particular, the network seems to exhibit structural compositionality for Contact, but not for Number, just like in the original experiment. We will be sure to add this experiment to the appendix!

---

### Official Review · Reviewer_xQUi · 2023-07-07

**Soundness:** 3 good
**Presentation:** 3 good
**Contribution:** 2 fair
**Rating:** 6
**Confidence:** 3

**Summary:**

The authors investigated to what extent the standard neural networks of the present day trained on the tasks solvable by composing subroutines result in modular structures reflecting the tasks' compositional nature (called structural compositionality in this study). To answer this question, they took the following approach on the trained networks: 1) use a model pruning technique to find the best subnetworks for individual subroutines, then 2) check the patterns in  accuracy difference of the discovered subnetworks and the ablated networks (complemental subnetworks with respect to the first ones) on the tasks corresponding to individual subroutines. They conducted experiments on both vision and language tasks with multiple neural networks and pretraining regimes. Based on the results, they concluded that the neural networks oftentimes exhibit structural compositionality even when there are no explicit inductive biases toward such solutions.

**Strengths:**

Making a learning machine that has compositional (systematic) generalization capability like a human is an important goal yet to be achieved in the field of artificial intelligence. On one hand, standard neural networks that are not explicitly imposed inductive biases for compositionality can generally be said to fail in compositional generalization as stated in lines 56 - 57, but on the other hand, these networks show compositional generalization capability to some extent in some cases, although not perfect and inferior to the ones that are imposed inductive biases. Revealing what is going on inside a neural network is an important research topic in this context, and the approach and results of this study will be of interest to the NeurIPS audience.

The idea of the approach took in this study is clear and reasonable. The model pruning method, experimental logic and concrete experiments are explained fairly well (some part is given in the Appendix).  The experiments are fairly rich in terms of tasks (both vision and language tasks were conducted and there are varieties in each) and models (ResNet50, WideResNet50, and ViT were tried for visual tasks, and BERT-small was tried for language tasks).


**Weaknesses:**

1. The URL of an Anonymous GitHub repository is provided in the paper (footnote on page 2, https://anonymous.4open.science/r/Compositional_Subnetworks-C8AB/). However, it is very hard to access the contents of the repo, because, although the repo consists of many subdirectories, there is no README in the top-level directory giving an overview.  This cast a shadow on the reproducibility.
1. Another relatively weak point of this paper is the contextualization relative to Csordás et al. 2021. In lines 49 and 259, Csordás et al. 2021 is explained as a work merely on a multitask setting, but it studied compositional (systematic) generalization settings using the SCAN (Lake and Baroni 2018) and the Mathematics Dataset (Saxton et al. 2019).



Csordás, R., van Steenkiste, S., and Schmidhuber, J. Are neural nets modular? Inspecting functional modularity through differentiable weight masks. In Proc. of the 9th International Conference on Learning Representations (ICLR), 2021.

Lake, B. M.  and Baroni, M. Generalization without systematicity: On the compositional skills of sequence-to-sequence recurrent networks. In Proc. of the 35th International Conference on Machine Learning (ICML), pp. 2873--2882, 2018.

Saxton, D., Grefenstette, E., Hill, F., and Kohli, . Analysing mathematical reasoning abilities of neural models. In Proc. of the 7th International Conference on Learning Representations (ICLR), 2019.

**Questions:**

Major Suggestions
1. I suggest adding README explaining the overall repository in the top-level directory during the Author Rebuttal period.
1. I suggest better contextualizing this work with respect to Csordás et al. 2021 in the paper.

Please also refer to Weaknesses section above.

Minor Sugestions
1. It would be better to mention WideResNet in Section 7 (Results) and point to Appendix.
1. It would be nice if applicability of the proposed approach to recurrent neural networks are explained in the paper.




**Limitations:**

The authors describes the limitations of their current method and the results reported in the paper in Section 10 (Discussion):
1. Their current method requires one to specify which subroutines to look for in advance.
1. Their current method requires one to use causal ablations and control models to properly interpret.
1. The reported results do not contain any analysis on the relationship between structural compositionality and compositional generalization.

\# Personally, I am very interested in 3, which is fully left for the future work.

---

> ### Author Rebuttal · Authors · 2023-08-09
>
> Thank you for your excellent feedback on our work! We have updated the README in the anonymous repository. We agree that it is valuable to add an extended discussion of Csordas et al. 2021 in this paper, and we will include a full paragraph in our related work section to elaborate on the relationship between our contribution and theirs. In particular, we will note how their work convincingly demonstrates that the same underlying algorithm is not being used in different data partitions, which implies that the network is behaving non-compositionally. One interesting difference between their study and ours is that our study explicitly looks for the compositional subroutines that a model might be implementing, whereas Csordas et al. looks for subnetworks that solve subsets of the dataset. Also, we will certainly comment on the WideResNet results in the main body, thanks for pointing that out!

---

> > ### Comment · Reviewer_xQUi · 2023-08-12
> >
> > Thank you very much for dealing with my suggestions.  Here are two follow-up comments.
> > * Regarding the major suggestion 2, can you show the draft of the paragraph regarding Csordas et al. 2021 here?
> > * Can you provide any comments about the applicability (extensibility) to recurrent networks? (Related to the minor suggestion 2. This is minor, and it is completely OK if it is currently unclear. Please just tell me so if it is the case. I'm just curious. )
> >
> > \# The major suggestion 1 has been completely addressed, thank you.

---

> > > ### Author Response · Authors · 2023-08-16
> > >
> > > No problem! Here is a draft of the Csordas et al. 2021 paragraph. We welcome any feedback that you might have on it!
> > >
> > > "Most directly related to the present study is Csordas et al. 2021, which also analyzes modularity within neural networks using learned binary masks. Their study also finds evidence of modular subnetworks within a multitask network: Within a network trained to perform addition and multiplication, different subnetworks arise for each operation. Csordas et al. 2021 also investigates whether the subnetworks are reused in a variety of contexts, and find that they are not. In particular, they demonstrate that subnetworks that solve particular partitions of the SCAN or Mathematics dataset oftentimes do not generalize to other partitions. From this, they conclude that neural networks do not flexibly combine subroutines in a manner that would enable full compositional generalization.
> > >
> > > However, their work did not attempt to uncover subnetworks that implement specific compositional subroutines within these compositional tasks. For example, they did not attempt to find a subnetwork that implements the "repeat" operation for SCAN, transforming "jump twice" into "JUMP JUMP".  Our work does attempt to find such compositional subroutines (e.g. a subroutine that implements "inside" or "contact"), and finds that these subroutines are often represented by modular subnetworks. This finding extends Csordas et al.'s result on a simple multitask setting to more complex compositional vision and language settings, and probes for subroutines that represent intermediate computations in a compositional task (i.e. "inside" is a constituent computation when computing "Inside-Contact"), rather than full solutions to particular tasks in a multitask setting (i.e. the "addition" subroutine provides a complete answer when the input specifies that the network must perform addition)."
> > >
> > >
> > > With respect to your question about recurrent networks: It is currently unclear to us whether recurrent networks would exhibit more/less/the same structural compositionality as feedforward networks! This is a very interesting question, and we will note it as an exciting direction for future work.

---

> > > > ### Comment · Reviewer_xQUi · 2023-08-18
> > > >
> > > > Thank you very much for the additional answers. All of my suggestions and questions have been adequately responded. I raised the score for Presentation from 2 to 3 and that for Rating from 5 to 6.

---

### Decision · Program_Chairs · 2023-09-21

**Decision:**

Accept (spotlight)

**Comment:**

I will recommend this paper for acceptance, because the reviewers were very positive about the work, stating that the approach is good and the paper is interesting (xQUi, W9E9, oWEk, DSTn), explained well (xQUi, W9E9, DSTn), and that the experiments were rich in tasks (xQUi, W9E9), and in models (xQUi).

Rebuttal was active, informative, and cordial. New information from authors was appreciated by reviewers, and some raised their scores as a result.

Some reviews pointed out some weak points of the work, including that it was hard to know if the results are generally true, or just true of the existing models (W9E9). Other reviewers mentioned that the approach requires that one knows what to look for (W9E9). While I believe these are reasonable points to bring up, they are not unique weakpoints of this paper and the benefits of this work suggest it reaches the bar for me.

For any future version of the work, I would make the following recommendations based on the reviewers' suggestions and my own understanding of the work: improve repo to enable reproducibility as suggested by (xQUi), better contextualization with related work (xQUi, DSTn). If possible DSTn’s limitation #1 could be considered/explored.